# Cell type-specific transcriptional programs in mouse prefrontal cortex during adolescence and addiction

Aritra Bhattacherjee[1,2,3,5], Mohamed Nadhir Djekidel [1,2,3,5], Renchao Chen[1,2,3,5], Wenqiang Chen [1,2,3,5], Luis M. Tuesta[1,2,3] & Yi Zhang [1,2,3,4]

Coordinated activity-induced transcriptional changes across multiple neuron subtypes of the prefrontal cortex (PFC) play a pivotal role in encoding and regulating major cognitive behaviors. Yet, the specific transcriptional programs in each neuron subtype remain unknown. Using single-cell RNA sequencing (scRNA-seq), here we comprehensively classify all unique cell subtypes in the PFC. We analyze transcriptional dynamics of each cell subtype under a naturally adaptive and an induced condition. Adaptive changes during adolescence (between P21 and P60), a highly dynamic phase of postnatal neuroplasticity, profoundly impacted transcription in each neuron subtype, including cell type-specific regulation of genes implicated in major neuropsychiatric disorders. On the other hand, an induced plasticity evoked by chronic cocaine addiction resulted in progressive transcriptional changes in multiple neuron subtypes and became most pronounced upon prolonged drug withdrawal. Our findings lay a foundation for understanding cell type-specific postnatal transcriptional dynamics under normal PFC function and in neuropsychiatric disease states.

[1] Howard Hughes Medical Institute, Boston Children's Hospital, Boston, MA 02115, USA. [2] Program in Cellular and Molecular Medicine, Boston Children's Hospital, Boston, MA 02115, USA. [3] Division of Hematology/Oncology, Department of Pediatrics, Boston Children's Hospital, Boston, MA 02115, USA. [4] Department of Genetics, Harvard Medical School, Boston, MA 02115, USA. [5] These authors contributed equally: Aritra Bhattacherjee, Mohamed Nadhir Djekidel, Renchao Chen, Wenqiang Chen. Correspondence and requests for materials should be addressed to Y.Z. (email: yzhang@genetics.med.harvard.edu)

Interactions with the natural and social environment drive animal behavior, and thereby support survival. Such interactions are largely adaptive, adjusting not only with the continuously changing environment, but also progressively throughout the postnatal development of an organism. As the site of origin of executive function, the prefrontal cortex (PFC) is a major brain region regulating behavior[1]. The PFC encodes and regulates the highest cognitive functions (such as learning, memory, judgment, decision-making, emotion, risk assessment or social behavior) through dynamic integration of sensory-stimulus-driven inputs from multiple brain regions[2]. A myriad of neuronal cell types of the PFC participate in this process. To sustain this dynamic experience-dependent neuroplasticity, the PFC, unlike any other brain region, continues to develop significantly during postnatal life, and retains a certain degree of plasticity lifelong[3].

Neural activity can induce widespread transcriptional changes in neurons[4], which in turn can encrypt structural and/or functional changes in them[4]. This is a principal basis of experience-dependent neuroplasticity that elicits stable long-term neural circuit changes. Stimulus-driven plasticity in PFC proceeds through early life experiences to shape the behavioral circuits of an organism. However, profound sensory experiences even in later life can significantly impact circuits, which in turn can recondition behavior[5,6]. Dysfunctions of both the early and later life events have been associated with various cognitive and psychiatric disorders. For example, several genetic aberrations can affect early life PFC development leading to psychiatric disorders such as schizophrenia, bipolar disorder, chronic depression, mania, or personality disorders[7]. On the other hand, major emotional or psychological episodes like chronic stress, trauma, drug addiction can alter brain function and behavior even much later in life[5,6]. However, the cellular mechanisms underlying these dysfunctions are largely unknown.

Deciphering the coordinated transcriptional programs across the various neuron subtypes of PFC during adaptive (taking place normally through postnatal development) or induced (initiated by major later life events) plasticity is at the heart of understanding both the biology and pathology of cognitive behavior. However, the profound cellular heterogeneity of the PFC has hindered the study of cell-type-specific transcriptional dynamics. To overcome this technical barrier, we performed single cell RNA sequencing (scRNA-seq) to classify all neuron subtypes in mouse PFC. We then analyzed transcriptional dynamics in each of the neuron subtypes between adolescence[8,9] (P21) and adulthood (P60) in mouse, a critical postnatal period believed to be associated with greatly increased neuroplasticity, and manifestation of most neuropsychiatric disorders[7,10]. In parallel, to understand transcriptional changes during induced plasticity in the mature brain, we analyzed volitional drug-intake using the cocaine intravenous self-administration (IVSA) model[11], a gold standard in addiction studies.

Using the two models (adolescence and drug addiction), we uncovered transcriptional dynamics in each neuron subtype associated with adaptive and challenge-induced neuroplasticity, highlighting distinct and conserved mechanisms. Comparative analysis of the transcriptional dynamics of various neuropsychiatric disease-relevant genes in PFC neuron subtypes revealed cellular/neural bases of these disorders. Collectively, our study establishes a foundation for understanding postnatal transcriptional dynamics in PFC and its relationship to the various cognitive and psychiatric disorders.

## Results

**Cellular composition of PFC**. To understand the cellular heterogeneity of PFC, we sought to classify the cell types based on their transcriptome. To this end, the PFC was dissected from acute coronal brain sections of P60 mice (Fig. 1a). We isolated the region that is broadly accepted as the PFC which predominantly encompass the anterior cingulate, prelimbic and infralimbic areas. After the tissue was enzymatically dissociated into single cell suspension, the cells were captured with the 10X Chromium platform (10X Genomics, CA), and used to construct cDNA libraries for sequencing (Fig. 1a).

We sequenced a total of 29,864 single cells from 12 independent biological samples (Supplementary Fig. 1a, Supplementary Data1). We first filtered out cells with potential double droplets (characterized with an unusually high number of detected genes) and unhealthy cells that generally have high mitochondrial mRNA loads (>10%) (see methods). We found that different cell types express different number of genes, particularly between neuronal and non-neuronal cells (Supplementary Fig. 1b). Thus, we applied a slightly different criteria to filter out low quality neuronal and non-neuronal cells (<800 genes for non-neuronal and <1500 genes for neuronal cells detected per cell) (refer to Methods). In the end, we obtained 9195 high quality non-neuronal (median UMI: 3764; median 1658 genes/cell) and 15,627 neuronal (median UMI: 13,194; median 3914 genes/cell) cells (Supplementary Fig. 1c), which can be separated into 8 major cell clusters (Fig. 1b). Importantly, each of the 8 major cell clusters can be detected in each of the 12 samples (Supplementary Fig. 1d). Based on the expression of cell type-specific markers, the non-neuronal cells were clustered as: astrocytes ($Gja1^+$), oligodendrocyte ($Aspa^+$), newly formed oligodendrocytes ($Bmp4^+$), oligodendrocyte precursors ($Pdgfra^+$), microglia ($C1qa^+$) and endothelial cells ($Flt1^+$) (Fig. 1c, d). The neurons express $Snap25$ and can be divided into excitatory ($Slc17a7^+$) and inhibitory ($Gad2^+$) neurons (Fig. 1c, d). The excitatory neurons form the largest (52.3%) cell class in the PFC, while the inhibitory neurons comprise a smaller portion (4.3%) of the total populations, consistent with the general excitatory/inhibitory ratio reported in most cortical areas (Supplementary Data 2).

**Molecularly distinct neuronal subtypes in the PFC**. Neurons of the cortex are diverse, and have been classified by one or combinations of different criteria such as morphology, electrical properties, anatomical location and histological features[12]. Despite apparent similarity, distinct functional subpopulations are believed to exist within the same neuron types classified by above approaches. Since the structure, function or behavior of a cell must have an impression on its gene expression, a cell's transcriptome can be a unifying basis for its classification. Indeed, we found that excitatory and inhibitory neurons of the PFC exhibit great transcriptional differences, forming distinct cell subtypes.

Excitatory neurons are the largest cell population in the PFC and are comprised of subtypes projecting to intra-cortical or long-distance targets. A broad "low resolution" classification of excitatory neurons revealed 13 transcriptionally distinct clusters or subtypes (hereon referred as Exc-1, 2,..13) (Fig. 2a, Supplementary Fig. 2a). Each subtype can be identified by unique expression of one or combination of 3–4 markers (Fig. 2b). Unique markers projected on t-SNE plots showed selective expression in respective neuron populations (Supplementary Fig. 2b). RNA in situ hybridization confirmed the presence of transcriptionally and anatomically distinctive neuron subtypes revealed by scRNA-seq (Supplementary Fig. 2c). Multi-channel single molecule fluorescence in situ hybridization (smFISH) further confirmed the presence of very distinct non-overlapping neuron populations. Co-staining for unique markers detected the

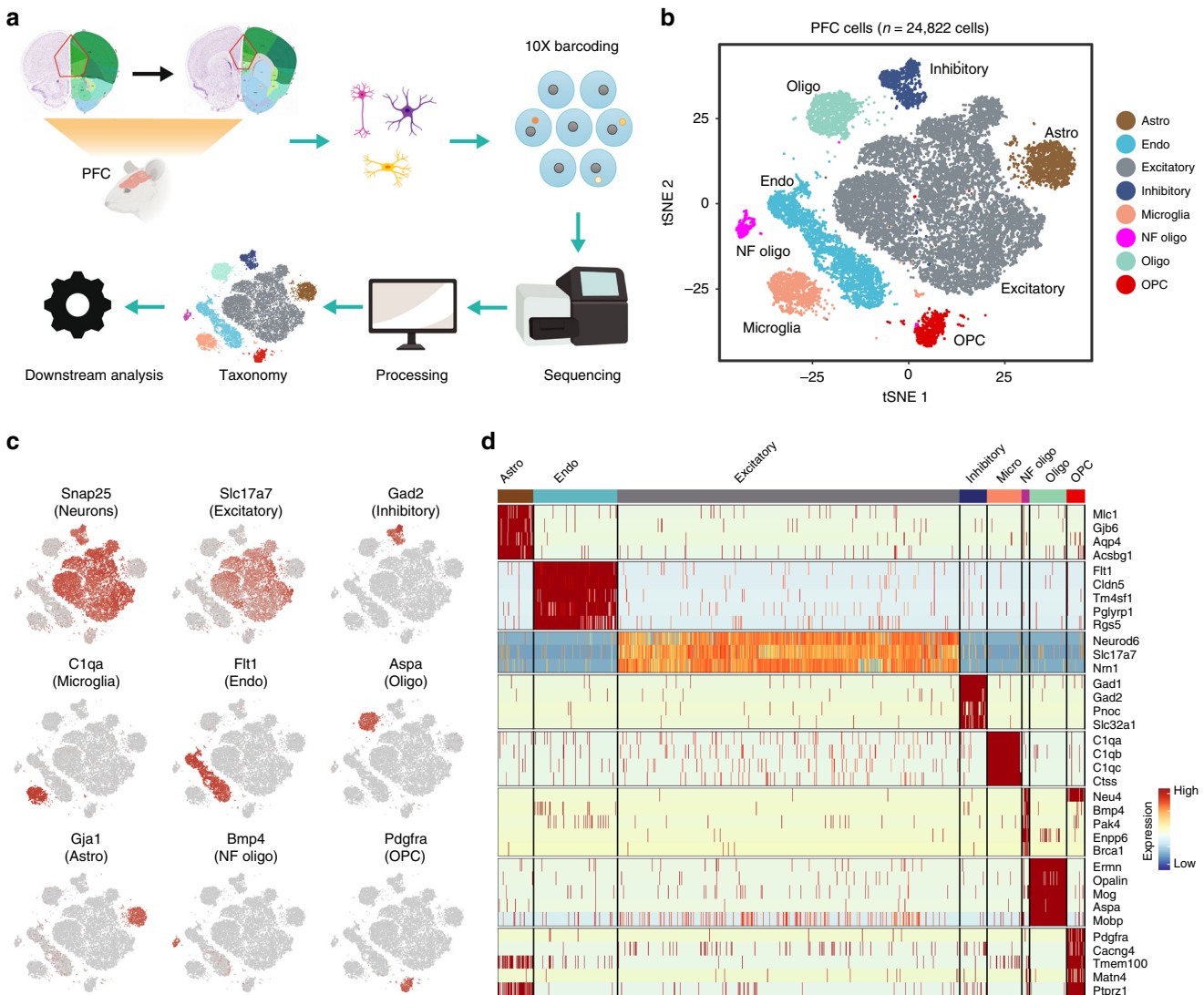

**Fig. 1** Transcriptome-based cell classification in mouse PFC. **a** Schematic of the experimental design. PFC sections (red outline) were isolated for scRNA-Seq. Created with BioRender.com. **b** t-SNE plot showing the broad clustering of PFC cell types based on transcriptome. **c** Expression of cell type-specific markers in each broad cell cluster color-highlighted on t-SNE plots. **d** Single cell heat map showing cell type-specific gene marker expression in the different PFC cell clusters

Exc-10 (*Pou3f1+*) and Exc-12 (*Tshz2+*) neurons adjacently within the same layer, while Exc-13 (*Foxp2+*) neuron cluster was detected mainly in a deeper layer (Fig. 2c).

Cortical excitatory neurons show remarkable laminar organization into histologically distinct layers which is tightly coupled to their projection, connections and function[13]. The PFC is comprised of 4 layers (L) L1, L2/3, L5, and L6 (but does not contain a L4). To detect the layer identity of the 13 excitatory neuron subtypes, we overlaid the expression of unique layer marker genes on the excitatory subpopulations (Supplementary Fig. 2d). For example, *Cux2* or *Calb1* represent L2/3 identity, *Etv1* is L5, while *Syt6* is in L6 (Supplementary Fig. 2e). The L1 is sparsely populated, containing few inhibitory neurons, not only in PFC, but throughout the entire cortex[14]. The comparative analysis revealed layer identity of each neuron subtype including several closely related subtypes in each layer e.g., L5-1, L5-2, and L5-3 within L5 (Fig. 2d, see the table in Supplementary Fig. 2d for a layer-cluster relationship). Multi-channel RNA-smFISH revealed that the distinct non-overlapping *Pou3f1+* (Exc-10) and *Tshz2+* (Exc-12) neurons both express *Etv-1*, revealing their L5 identity (Fig. 2e). Previous studies have demonstrated that

different PFC layers are involved in different neuronal functions[13]. For example, L2/3 are largely involved in intra-cortical (cortico-cortical) regulations, while L5 sends long distance projections outside cortex and L6 receives subcortical inputs (e.g., from thalamus) and projects to superficial layers[13]. Thus, layer properties confer further identity to each subtype, which provides a foundation for understanding cell cluster-specific function.

Like most other cortical areas, the inhibitory neurons are present in much lower numbers in the PFC, only accounting for 4.3% of the total PFC cells (at a ratio of 1:12 with excitatory neurons). Unlike excitatory neurons, inhibitory neurons do not show distinct laminar organization, although some types may be differentially enriched within particular cortical layers. Classification revealed 12 clusters (referred as Inhib-1, 2,..12), highlighting functional diversity and specialization (Fig. 2f, Supplementary Fig. 3a). As expected, the major known inhibitory neuron populations like *Sst+*, *Pavlb+* and *Vip+* were detected (Fig. 2g, Supplementary Fig. 3b) and could also be confirmed from RNA-FISH (Supplementary Fig. 3c). Developmentally, *Sst+* and *Pavlb+* originate from the medial ganglionic eminence (MGE), while the

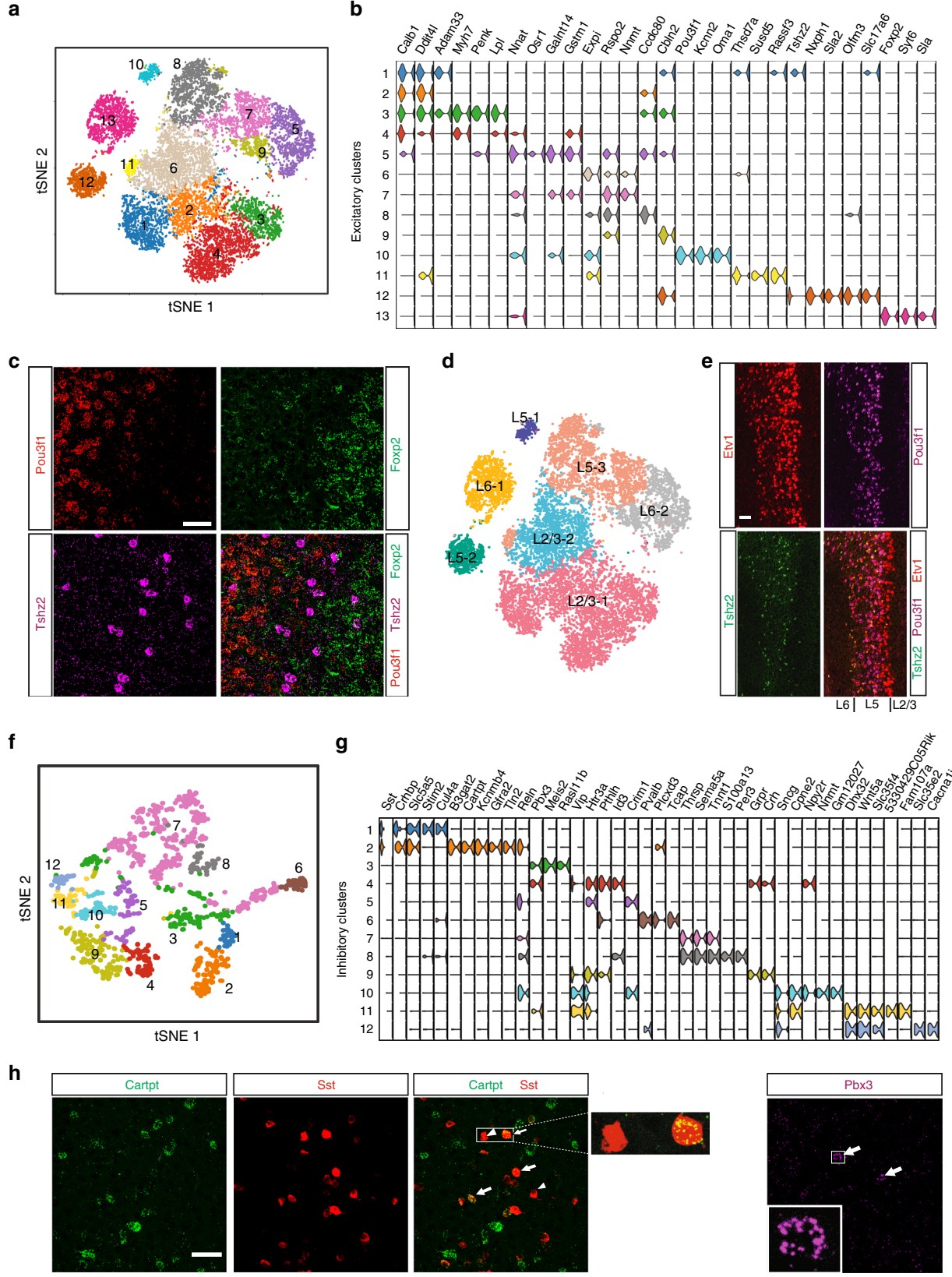

*Vip+* originate from the caudal ganglionic eminence (CGE)[15]. Based on the expression of combinational markers, these major inhibitory neuron populations can be further classified (Fig. 2g). For example, the *Sst* population can be further classified into two broad subtypes based on the expression of *Stim2* and *Cul4a* or

otherwise *B3gat2*, *Cartpt*, and *Kcnmb4* (Fig. 2g). By co-staining for *Cartpt* (*cocaine- and amphetamine-regulated transcript*) and *Sst* using smFISH, we confirmed the *Sst+*/*Cartpt+* and *Sst+*/*Cartpt−* cells belonging to distinct cell subtypes (Fig. 2h). Likewise, markers like *Reln* alone can discriminate subtypes

**Fig. 2** PFC contains distinct excitatory and inhibitory neuron subtypes. **a** t-SNE plot showing that excitatory neurons of PFC can be broadly classified into 13 unique subtypes based on their transcriptome. **b** Violin plot showing expression of specific markers for each of the 13 excitatory neuron clusters or subtypes. **c** Multi-channel FISH detecting distinct excitatory neuron types within the same cortical layer (*Pou3f1* and *Tshz2* in L5) or across different layers (*Foxp2* in L6). **d** Assigning excitatory neurons to respective cortical layers by projecting expression of layer-specific markers onto the t-SNE plot. **e** Identification of two distinct cell types (*Pou3f1*+ and *Tshz2*+) in L5 that commonly express L5 marker *Etv1*. **f** t-SNE plot showing classification of PFC inhibitory neurons into 12 distinct subtypes based on their transcriptome. **g** Violin plot showing distinct markers for each of the 12 different subtypes of inhibitory neurons. **h** Multi-channel FISH detecting subtypes within a known inhibitory neuron subpopulation (*Cartpt*+, arrow, and *Cartpt*−, arrowhead within *Sst* neurons: enlarged view of single cells in box area shown in side panel) and a rare neuron subtypes (*Pbx3*+: inset is an enlarged view of single cell)

within more than one of the major inhibitory neurons, such as Inhib-2 (*Sst*+/*Reln*+) vs Inhib-1 (*Sst*+/*Reln*−), or Inhib-10 (*Vip*+/*Reln*+) vs. Inhib-9 (*Vip*+/*Reln*−) (Fig. 2g). Projection of *Reln*, *Sst* or *Vip* expression on the inhibitory t-SNE clearly showed overlap on the expression of these markers in certain inhibitory neurons (Supplementary Fig. 3b). Indeed, broad expression of *Reln* in coronal sections within PFC can be confirmed from RNA-FISH (Supplementary Fig. 3c). Such transcriptional diversity of specific neuron subtypes may reflect functional specialization.

Importantly, we detected at least four distinct inhibitory subpopulations i.e., Inhib-3 (*Meis2*+ or *Pbx3*+), Inhib-7, 8 (*Tnnt1*+, with or without *Id3*+) and Inhib-12 (*Slc35e2*+, *Cacna1i*+), which do not belong to any of the principle *Pvalb*, *Sst* or *Vip* populations, and are likely the remaining non-MGE origin cells (arising from CGE and/or POA)[14–16] (Fig. 2g, Supplementary Figs. 3b, 3c). The rare *Pbx3*+ neurons could be clearly detected using smFISH (Fig. 2h). Some non-inhibitory neuron populations also appear to express *Meis2* (Supplementary Fig. 3c), making *Pbx3* a more selective marker. While divergent inhibitory neuron populations have been reported in other cortical regions based on different marker combinations[14], we detect the distinct populations in PFC and define characteristic regional markers for their identification.

In summary, using scRNA-seq, we revealed the comprehensive neuronal subtypes comprising the anatomically defined PFC region and their corresponding cortical layer identity. This finally also provides us the basis for understanding neuron subtype-specific transcriptional programs regulating PFC function.

**PFC transcriptional features are different from VISp or ALM**. To further elucidate unique transcriptional features of the PFC, we compared excitatory neuronal clusters of the PFC to those of other cortical regions, the primary visual cortex (VISp) and anterolateral motor cortex (ALM). To enable a better comparison, we first performed a "higher resolution" clustering of the PFC excitatory neurons, which identified 26 distinct subtypes (Fig. 3a), a number comparable to the excitatory neuron subtypes of VISp and ALM reported by Tasic et al.[14]. Unique markers could be detected for each of the 26 subtypes (Fig. 3b) that showed specific enrichment on the t-SNE plots (Supplementary Fig. 4a) and could be validated histologically from RNA-FISH (Supplementary Fig. 4b).

To identify transcriptionally similar excitatory neuron subtypes between these regions, we calculated similarity of PFC with VISp and ALM, respectively, in the aligned canonical correlation space (CCA)[17] to limit the batch effect, followed by a neighbor voting strategy as implemented by the MetaNeighbor method[18]. This generated a similarity map of the different neuron subtypes. Similar neuron subtypes 'grouped' together with PFC versus VISp (G1-G16, green blocks on top heat map Fig. 3c) and ALM (G1-G13, green blocks on bottom heat map Fig. 3c). While most excitatory neuron subtypes of PFC have corresponding cell clusters in VISp or ALM group together, clusters that are unique to PFC do exist. For example, PFC-9 (G13) or PFC6,15,16,17

(G11) showed no similarity to any VISp subpopulations, although they grouped with L5-IT projection cells in the ALM (Fig. 3c: PFC labeled in blue, VISp red and ALM purple). *Rspo2*, a common marker for the G11 members, showed distinctly enriched expression in PFC, but no appreciable signal in VISp (Fig S4c). Similarly, PFC clusters 7 and 14 (G7) showed no similarity to the ALM subpopulations but resembled the L6IT-5 (Fig. 3c, Supplementary Data 3). Accordingly, *Nnat*, a selective marker for G7 is seen enriched in PFC but depleted in ALM (Supplementary Fig. 4d). Likewise, there were some unique clusters in both VISp and ALM that corresponded to no cluster of the PFC, for example, CR1of ALM, and L6IT-2 and L5PT-1 of VISp (Fig. 3c, Supplementary Data 3)[14].

We next compared gene expression between neurons of PFC vs. VISp or ALM within each similar group. We found that despite the close resemblance, significant transcriptional differences still exist among the clusters that share the same group. We found many genes (273–555) are differentially expressed between PFC and VISp or PFC and ALM (FC > 2, *q*-value < 0.05, likelihood test for zero-inflated data, Bonferroni corrected) (Supplementary Fig. 4e), further highlighting the transcriptional distinction of the excitatory neurons of the PFC. These results indicate that PFC has neuron populations that are transcriptionally distinct from its closest (ALM), as well as farthest (VISp) neighbors, which may explain unique functions of the PFC compared to the other cortical regions.

Considering the marked differences, especially between VISp and PFC, we further asked whether the G11 neurons (Fig. 3c) of the PFC exclusively express genes that are functionally relevant only in that area. Comparison of all VISp neurons with G11 revealed that upto 30 molecules are highly expressed in G11 of PFC but depleted across all VISp (Supplementary Data 3). These are predominated by receptors, ion-channels, transporters etc. Interestingly, G11 of PFC showed enrichment of molecules like *Scn7a*, a voltage gated sodium channel (implicated in epilepsy) or *Cacna1i*, a low voltage calcium channel (implicated in schizophrenia) that are uniquely relevant to cognitive function as opposed to visual processing (Supplementary Data 3).

Overall, these analyses revealed that PFC neurons have very distinct transcriptional properties relative to other cortical regions, which may underlie their unique functional abilities for dynamic integration of varied convergent signals, or their sustained postnatal neuroplasticity/adaptability.

**Cell type-specific transcriptional changes during adolescence**. Adolescence marks one of the most dynamic phases of experience-dependent plasticity in the PFC[8]. Mice are weaned from maternal care at P21 and begin exploring the environment on their own, and are considered as adult at P60. To characterize the transcriptional changes associated with this period, we performed scRNA-seq of P21 mouse PFC. Following the same filtering criteria of P60 PFC, we obtained 10,646 high quality (median of 2808 genes and 7384 UMI per-cell) mouse P21 PFC cells.

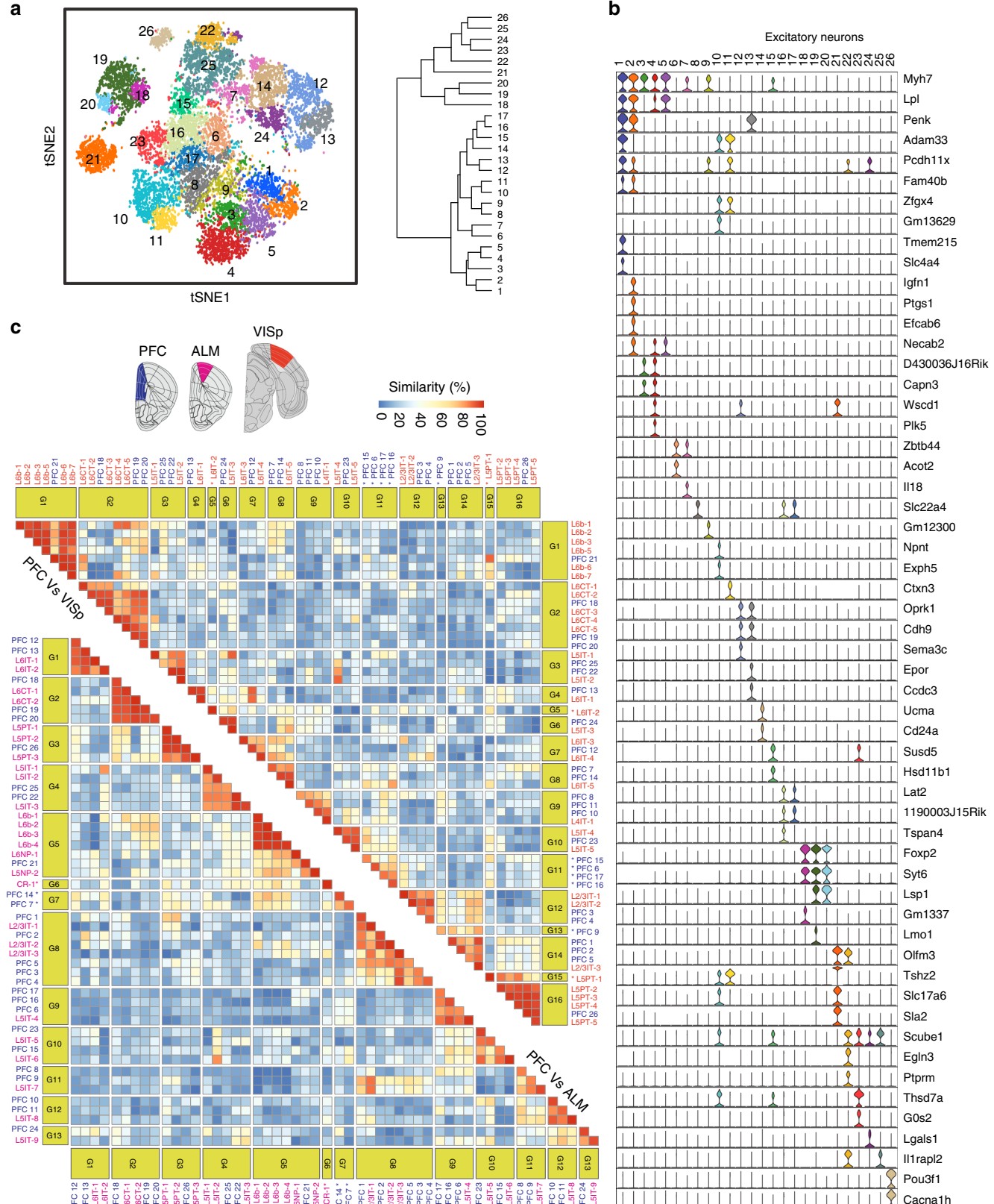

**Fig. 3** Distinct transcriptional features of PFC excitatory neurons relative to those of VISp and ALM. **a** t-SNE plot (left) showing high resolution clustering of PFC excitatory neurons and the dendrogram of hierarchical clustering (right). **b** Violin plot showing the distinct markers for each of the clusters in **a**. **c** Heat map showing the degree of similarity [0–100] between the excitatory neuron subtypes in PFC, primary visual cortex (VISp), and anterolateral motor (ALM) based on similarity of transcriptional profiles. Subtypes showing a similarity ≥ 90% were clustered together into blocks. Subpopulations showing <90% similarity to any of the subpopulation from the other dataset were considered as region specific (shown with bold font and *). VISp and ALM cluster names were abbreviated, the original corresponding names are available in Supplementary Data 3

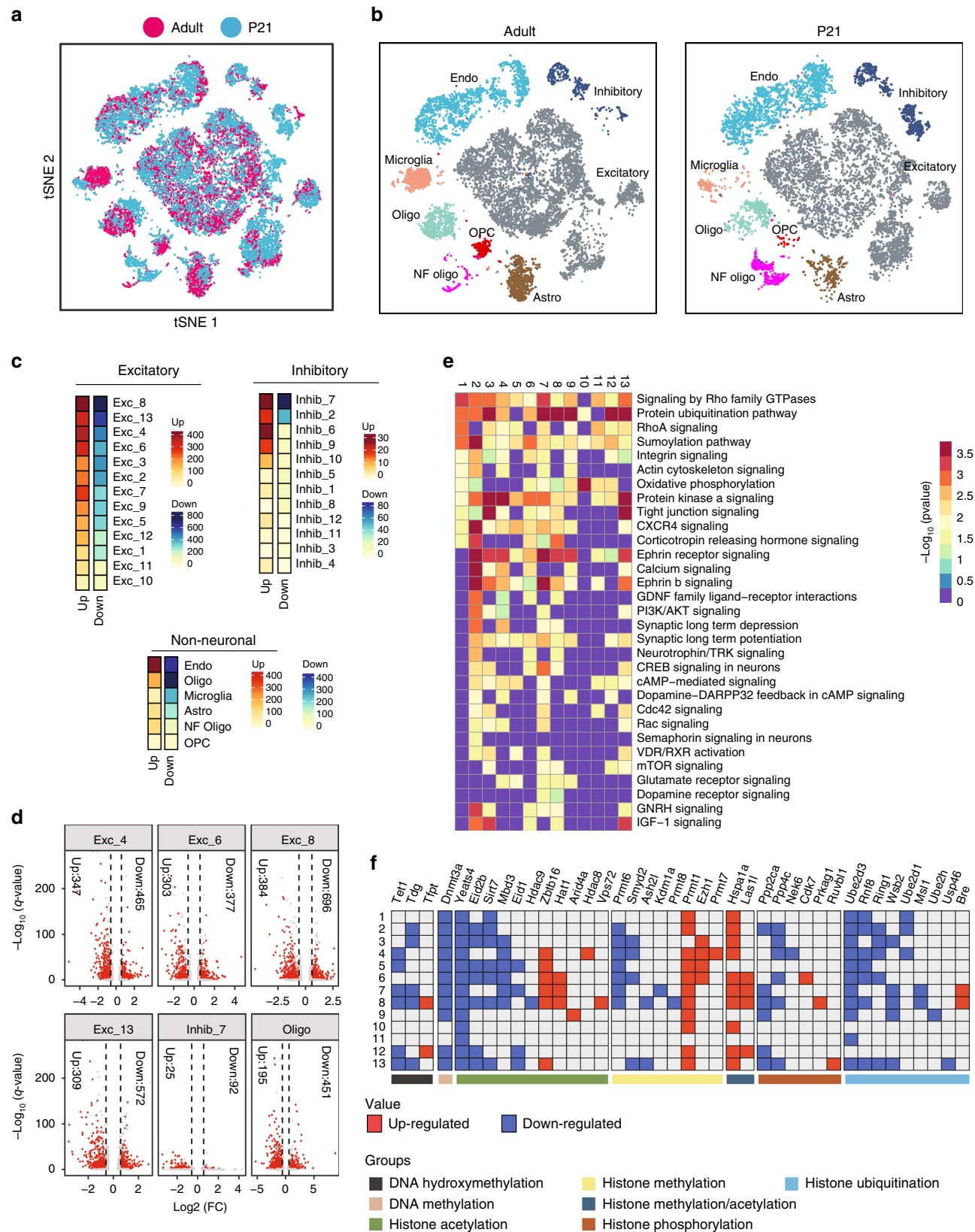

To detect similar populations and identify corresponding cell clusters between the 10,646 P21 cells and the 11,886 P60 PFC cells, we aligned the two scRNA-seq data sets in t-SNE by cross-correlation analysis (CCA)[17] (Fig. 4a). Using bootstrapped correlation, all clusters identified in the adult PFC are detected in the P21 PFC (Fig. 4b, see methods), consistent with prior

knowledge that formation and migration of different cell types in cortex is complete by this time[19].

Comparison of gene expression in PFC of P21 with that of P60 mouse revealed dramatic changes across all cell subtypes (Supplementary Data 4). With a cutoff of 0.05 FDR and at least 50% change, expression (negative binomial generalized linear

**Fig. 4** Widespread transcriptional changes in PFC cell types between P21 and P60 mice. **a** t-SNE plot showing the distribution of the merged cells from P60 (purple) and P21 (light-blue) PFC cells after alignment using correlation analysis (CCA). **b** t-SNE plot showing the cell-type assignment and proportion of the cell-types in the P60 (left) and P21 (right) PFC cells. **c** Number of genes dynamically changed in excitatory neurons, inhibitory neurons and non-neuronal cells. Red: upregulated genes; Blue: downregulated genes. **d** Representative volcano plots showing altered gene expression in the indicated clusters (cutoff: FC > 1.5 and q-value < 0.05). **e** Heatmap showing the $-\log_{10}$(p-value) (right-tailed Fisher exact test) of the functional pathways impacted in excitatory neuron subtypes based on gene expression changes between P21 and P60. **f** Gene names and functional categories of epigenetic modifiers up-regulated (red) and down-regulated (blue) in each PFC excitatory neuron subtypes between P21 and P60 (cutoff: FC > 1.5 and q-value < 0.05; numbers 1–13 on left axis indicate the clusters). The differential gene expression p-values were calculated using the negative binomial generalized linear model and the q-values were derived using the Bonferroni method

model, Bonferroni corrected) of a minimum of 338 genes (Exc-1) and maximum of 1080 genes (Exc-8) were altered across the excitatory clusters (Fig. 4c). A significant number of genes were both up and down regulated in each cluster (Fig. 4c). Among the excitatory neurons, clusters Exc-4, Exc-6, Exc-8, and Exc-13 showed maximum transcriptional changes (Fig. 4c, d). To validate the scRNA-seq results, we performed smFISH which confirmed the cell type-specific transcriptional changes. For example, *Marcksl1* is greatly downregulated in P60 in Exc-12 and Exc-13 that are respectively marked by *Tshz2* and *Foxp2* (Supplementary Fig. 5a–c); *Ptgds* is upregulated in P60 in Exc-12 marked by *Tshz2* (Supplementary Fig. 5d); and *Hspa1a* is upregulated in P60 in Exc-10 and Exc-13 that are respectively marked by *Pou3f1* and *Foxp2* (Supplementary Fig. 5e, f).

Notably, Exc-4 (*Calb1*[+]/*Cux2*[+]) and Exc-6 (*Calb1*[−]/*Cux2*[+]) belong to the superficial layer L2/3. On the other hand, cluster Exc-8 and Exc-13 belong to L5 and L6, respectively (Supplementary Fig. 2d-table). Therefore, while transcription is significantly altered in all excitatory neurons, different subtypes in each layer are impacted to a different degree (Fig. 4c, d, Supplementary Fig. 6a).

Using the same cutoffs, differentially expressed genes, ranging from 6 (Inhib-3) to 117 (Inhib-7), were also detected across inhibitory neurons (Fig. 4c, d). The low abundance of inhibitory neurons can limit statistical power, resulting in a lower overall number of differential genes detected. However, a cut off of 1.5-fold change would incorporate most relevant genes (Fig. 4c, d). Interestingly, even the non-neuronal cells undergo robust changes in transcription, with the highest impact on endothelial cells with 879 genes affected (Fig. 4c). In addition, as many as 646 genes in mature oligodendrocytes, 288 genes in microglia, 268 genes in astrocytes, 209 genes in newly formed oligodendrocytes, and 53 genes in oligodendrocyte precursor were affected (Fig. 4c).

To explore how these transcriptional changes may affect the function of the various cell types, we performed a functional enrichment analysis on the differentially expressed genes. We found that within each cell subtype, transcriptional changes impacted different functional categories of molecules including enzymes, transcription factors, translation regulators, ion channels, GPCRs, cytokines, transporters, kinases, and phosphatases (Supplementary Fig. 6b). Interestingly, across cell types, the highest percentage of affected genes comprised enzymes (~50%), followed by transcription factors (~12–13%) and transporters (Supplementary Fig. 6b).

To further characterize the functional pathways affected by differential gene expression (p-value < 0.05, right-tailed Fisher exact test), we focused on excitatory neurons, the largest and most diverse cell type in PFC, and performed analysis using IPA (Ingenuity Pathway Analysis, Qiagen Bioinformatics). Pro-nounced impacts were detected on pathways related to actin cytoskeleton, membrane signaling, adhesion and tight junctions (Fig. 4e). Across all excitatory clusters, Rho signaling (affecting Rho GTPases, kinases etc.)[20] was one of the most strongly affected. In addition, ephrin, integrin, semaphoring, Rac, and

various growth factor signaling pathways which can impact synapse and axon growth were also strongly affected in certain cell clusters (Fig. 4e)[20–23]. Cumulatively, these pathways facilitate functions like membrane protrusion, ruffling, adherence, and neurite/dendrite growth that are pivotal for structural changes in synapses, dendrites and axons: implying extensive structural plasticity (or modifications) in most excitatory populations[20–24]. Widespread activation of ubiquitination or sumoylation, accompanied by CREB, PKA, cAMP or calcium signaling factors in several clusters is consistent with high transcriptional activity (Fig. 4e)[25–27]. Also, metabolic processes, like oxygen consumption and mitochondrial function are broadly modulated to keep pace with both development and plasticity, consistent with prior reports in neurodevelopment[28] (Fig. 4e). Not surprisingly, all these functions together contribute to long term potentiation or synaptic plasticity in many neuron subtypes (Fig. 4e), consistent with the hypothesis of increased neuroplasticity during adolescence.

Besides the widely regulated pathways, some pathways are selectively enriched in specific clusters. For example, corticotrophin releasing hormone signaling (known to alter pyramidal neuron excitability[29], Exc-1,2,6,7,8) dopamine receptor signaling (Exc-7,8), glutamate receptor signaling (Exc-4,5,7,8,9), CXCR4 signaling (Exc-1 to 9, 13) or IGF-1 signaling (Exc-2,3,6,8,13) selectively impacted only the relevant clusters (Fig. 4e).

Collectively, our comparative transcriptome analysis suggests that selective transcriptional programs operate in each PFC cell subtype during adolescence, driving some broad and some highly specific pathways that facilitate the adaptive transition of the PFC, which likely contribute to proper behavioral programing.

**Changes in epigenetic factors during adolescence.** Widespread change in transcription is often associated with broad chromatin and epigenetic regulations. Of the 720 known epigenetic modulators[30], we found that expression of many genes involved in histone modifications is altered when compared between P21 and P60 PFC (Fig. 4f: shows some prominent examples). For example, within the excitatory neurons, between 4 (in Exc-10) to 50 (in Exc-8) epigenetic regulators were significantly altered in the various subtypes, with most subtypes having 20–30 epigenetic regulators changed (Supplementary Data 5).

The altered histone modifiers can regulate gene expression either positively or negatively. For example, *Prmt1*, a histone arginine methylase associated with gene activation[31], is upregulated in most excitatory neuron subtypes; while *Yeats4* and *Ube2d3* involved in histone acetylation and ubiquitylation, respectively, are downregulated across most excitatory neuron subtypes (Fig. 4f). While others like *Arid4a*, *Hdac8*, and *Ube2h* are only altered in a specific neuron subtypes (Fig. 4f).

We realized that many of these altered genes are members of the 68 known epigenetic regulatory complexes[30], which have well-defined roles in transcription activation or repression. We identified the complexes to which these genes belong and

estimated the number of up- or down-regulated genes associated with each complex (Supplementary Fig. 6c). The major complexes identified include Chd8, NuRD, NuA4 or PRC2 (with one or more regulated members) (Supplementary Fig. 6c). For example, Hspa1a[32], a member of the CHD8 complex is upregulated in almost all excitatory neurons except Exc-5, 9, and 11 (Fig. 4f). Transcriptional regulation by Chd8 has been associated with neuronal function. Dysfunction of this complex has been associated with autism[33], and H4K16 acetylation by this complex is associated with brain aging[34]. When queried for Chd8 direct targets based on ChIP-seq results[35], we found multiple Chd8 targets are indeed regulated selectively in the different clusters (Exc-1, 2, 3, 4, 8, 13) (Supplementary Fig. 6d, Supplementary Data 5). Likewise, downregulation of Mbd3 (Fig. 4f), a component of the NuRD complex (Supplementary Fig. 6c), can prevent deacetylation to promote transcription[36]. Conversely, Yeats4 of NuA4 complex is downregulated across all excitatory neurons (Fig. 4f, Supplementary Fig. 6c), indicating repression[37]. Similarly, increased Ezh1 (of PRC2) (Fig. 4f) can promote repression by increasing H3K27 methylation, which has been associated with important neuronal functions[38,39].

Therefore, consistent with the large number of genes both up- and down-regulated (Supplementary Fig. 6a), we detected changes in both activating and repressive epigenetic regulators in each neuron subtype. Dnmat3a is widely downregulated across excitatory subtypes, and Tet1 involved in DNA demethylation is also downregulated in several clusters (Exc-4,5,7,8,12,13) (Fig. 4f). This is consistent with prior reports that DNA methylation events in cortex occur earlier in development[40]. It reinstates that epigenetic regulation during adolescence is likely achieved mainly through histone modifications as opposed to DNA methylation.

**Selective expression of neuropsychiatric disease genes.** Many neuropsychiatric disorders have been linked to genetic mutations and impaired gene expression[7]. The PFC is believed to play a central role in the pathophysiology of several neuropsychiatric disorders, including schizophrenia, bipolar disorder, psychosis, mania, depression or suicidal tendencies. Since initial manifestation of these disorders frequently occur during adolescence, transcriptional plasticity during this period is believed to play a key role in disease pathogenesis[10]. Genome wide association studies (GWAS) have revealed candidate genes whose mutations have been associated with these disorders[41]. However, the cellular and molecular mechanisms underlying their involvement remain largely unknown.

To understand the cellular mechanism, we asked whether these disease-relevant candidate genes exhibit cell type-specific expression in the PFC. To this end, we analyzed 12 neuropsychiatric disorders that have been directly or indirectly linked to PFC function. The disorders include schizophrenia, bipolar disorder, epilepsy, depression, personality disorder, obsessive compulsive disorder (OCD), autism, ADHD, alcoholism, suicide risk, mania, and dementia-Alzheimer disorder. The GWAS candidate genes were derived from the EMBL GWAS catalog (https://www.ebi.ac.uk/gwas/), and only those with reported exonic mutations were included in the analysis.

Using the transcriptomic data from the P60 mouse, we calculated expression enrichment across all the PFC cell clusters and identified that several disease-relevant genes (GWAS candidates) exhibit highly selective or enriched expression in specific cell types or subtypes of the PFC (Supplementary Data 6). For example, selective expression of schizophrenia candidates Sl17a6 and Lypd6 can be clearly visualized on the global PFC t-SNE (Fig. 5a, top 2 panels; cluster identity: Supplementary Fig. 8a), as well as on the excitatory and inhibitory t-SNE plots (Fig.

5a-left column, bottom 2 panels). Expression of Sl17a6 and Lypd6 in PFC can be verified from RNA in situ hybridization of coronal brain sections (Fig. 5a-right column, bottom 2 panels). Similarly, many schizophrenia-relevant genes exhibit selective expression in distinct neuronal and non-neuronal subtypes (Fig. 5c-top). In addition, we found that some schizophrenia-relevant genes are broadly expressed in a particular cell type. For example, Tbxas1 is expressed in all microglia, while Kif5c is expressed in all neurons (Fig. 5a, Supplementary Fig. 7). Furthermore, some other schizophrenia-relevant genes are expressed across all cell types (Supplementary Fig. 7, Supplementary Data 6). These broadly expressed genes often comprise enzymes, metabolic or cytoskeletal regulators essential for general cellular functions. Similar expression patterns on genes relevant to bipolar disorder, another leading PFC-related neuropsychiatric condition, can also be observed (Fig. 5b, c-bottom, Supplementary Fig. 7). Likewise, specific enrichment/expression patterns of GWAS candidates can be detected in every PFC-related disorder, with depression presented as yet another examples (Supplementary Fig. 8a, b). Similar results for each of the 12 disorders can be browsed in full details in our PFC explorer (https://www.zhanglab.tch.harvard.edu/neuro-group/PFCExplorer) (Supplementary Data 6). With the exception of Epilepsy and OCD, for all other tested disorders at least 25% of the GWAS candidate genes exhibit enriched expression in a particular cell subtype or cell type (i.e., neuron, astrocyte etc.) (Supplementary Fig. 8c).

To identify the most relevant cell cluster affected in a particular disorder, we calculated the percentage of the GWAS candidate genes expressed in each cell cluster in a particular disorder (Fig. 5d). For example, in schizophrenia, we found that excitatory neurons are highly affected, with Exc-9 and 11 as the most relevant cells as these cell clusters express the highest number of schizophrenia relevant genes (Fig. 5d). This analysis allowed us to identify the most relevant (frequently affected) cell type(s) for each disorder (Fig. 5d).

However, in the context of transcriptional regulation, one of the most important questions is whether expression of a GWAS candidate gene is regulated during adolescence, the time of onset of many neuropsychiatric disorders. To this end, we compared the expression of all GWAS candidates in each PFC cell subtype between P21 and P60 mice. We found that many candidate genes undergo significant change in expression during this period. For each disorder, the relative impact of these changes in different cell subtypes is best illustrated by projecting the number of candidate genes impacted in each subtype on the global PFC t-SNE. Accordingly, projection on the global t-SNE (Supplementary Fig. 8a) shows that for schizophrenia, Exc-13 accounts for the maximum number of differential GWAS candidate genes among the neurons, and the endothelial cells account for the maximum of the non-neuronal cells (Fig. 5e). Conversely, for bipolar disorder Exc-8 accounts for the most candidates (followed by Exc-13), while endothelial cells for the non-neuronal cells (Fig. 5e). Likewise, the cell type-specific differentially regulated GWAS can be identified in each of the 12 disorders we studied (Supplementary Fig. 8d). Remarkably, 26–36% of all GWAS candidates are dynamically regulated during adolescence for each disorder (Supplementary Fig. 8d). This analysis also highlights the subtypes that have the most differentially expressed GWAS candidate genes, hinting its importance in the disease biology. The analysis also reveals that many candidates in non-neuronal cells are also regulated during adolescence (Supplementary Fig. 8d).

The observation that GWAS candidate genes exhibit cell subtype-specific transcriptional dynamics has immediate and profound translational implications. For example, schizophrenia and bipolar disorder, two major PFC-related disorders, share

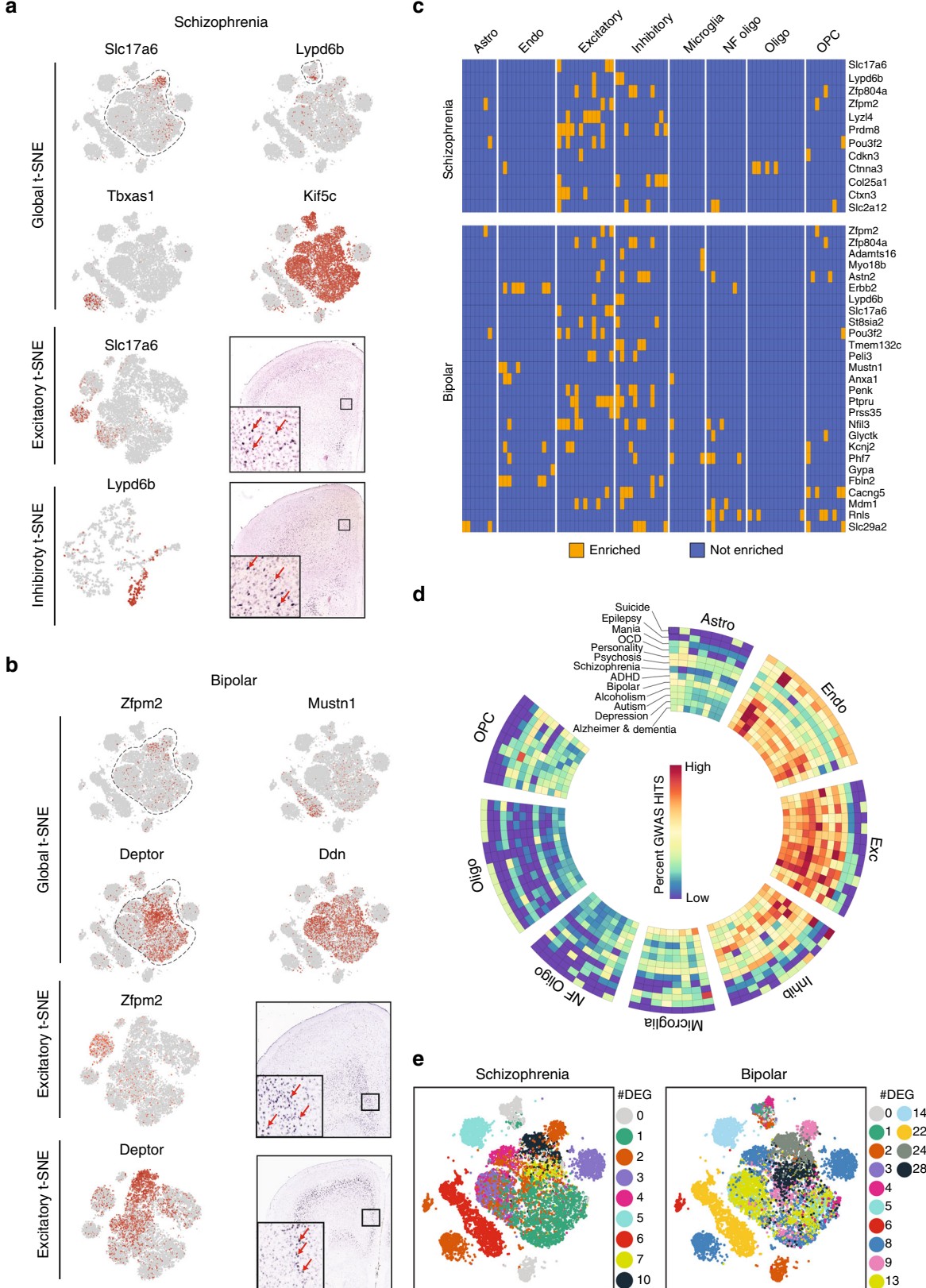

highly similar behavioral features and are often difficult to diagnose or treat differentially[42]. With our approach, many relevant genes exhibit cell type-specific expression for each disease, providing higher specificity for both diagnosis and treatment. For example, genes like *Adamts16*, *Myo18b*, *Astn2*, and *Erbb2* exhibit cell cluster-specific or enriched expression (Fig. 5c) and are only associated with bipolar, while some other genes, like *Zfp804a* and *Zfpm2*, are associated with both. Furthermore, across all disorders, better prognosis can be achieved based not only on cell type-enriched expression of the GWAS candidate, but also its differential regulation. Such approach can help in understanding disease mechanisms and cell type-specific treatments.

**Fig. 5** Expression of many GWAS candidates is enriched in specific clusters and dynamically changed during adolescence. **a** t-SNE plot showing examples of some cell type- and subtype-specific expression of GWAS candidate genes in schizophrenia. Top-panel shows the gene expression enrichment of four representative candidates on the global t-SNE plot. The bottom panels show zoomed t-SNE of excitatory and inhibitory clusters, respectively, and Allen Brain ISH images showing the specific gene expression. **b** Similar representations for bipolar disorder as shown for schizophrenia in **a**. **c** Heatmap showing GWAS candidate disease-relevant genes with cell subtype-specific expression. The blue color indicates no enrichment, the orange color indicates gene enrichment. Schizophrenia and Bipolar disorders are shown as examples. **d** Circular heatmap showing the percent of enriched subtype- and cell type-specific GWAS candidate genes in each of the 12 PFC-relevant diseases. Each disease is scaled individually, bright red: high enrichment, dark blue: depletion. **e** t-SNE-plot indicating the number of differentially expressed schizophrenia and bipolar GWAS candidates per cluster (shown in distinct colors) between P21 and P60 in each cell subtypes

**Cocaine induced cell-type specific transcriptional changes.** While disruption of adaptive neuroplasticity in adolescence can evoke various neuropsychiatric disorders, induced plasticity under psychological, emotional or stressful conditions like PTSD, depression, social isolation or drug addiction is believed to elicit significant cellular re-adaptations in PFC leading to long term behavioral changes[5]. Therefore, decoding gene expression programs associated with induced plasticity is another key component in understanding transcriptional plasticity that modulate PFC function. Psychological crises associated with induced plasticity are more common in adult life and present major social and clinical challenges. To address this question, we chose cocaine self-administration[11], a well-established drug addiction model.

To this end, C57BL/6 mice at P60 were implanted with intrajugular catheter connected to a pump to enable intravenous self-administration (IVSA). Mice were trained initially with food reward and then to intravenously self-administer cocaine by lever pressing (Fig. 6a). The experimental groups consisted of saline and cocaine IVSA (See Methods for details). The mice showed the expected behavioral patterns in IVSA reaching a steady level of cocaine acquisition at the maintenance phase (Fig. 6b). Animals were put on maintenance for 15 days and then cocaine was withdrawn. PFCs were then harvested from cohorts at the end of 15-day maintenance, at 48 h and 15-days withdrawal, respectively, for scRNA-seq (Fig. 6a). Comparable cell numbers and data quality were obtained from both saline (11,886 cells) and cocaine (12,936 cells) groups (Supplementary Figure 9a, b). Similar cell composition and cell clusters could also be detected in both saline and cocaine groups enabling comparative analysis of differential gene expression (Fig. 6c).

Unlike global adaptations that may occur in all cells of a neuron subtype during development, induced plasticity should mainly impact neurons within the circuits that respond to the specific stimulus. Thus to measure differential gene expression, we adopted the recently developed SC2P method which accesses transcriptional regulation in terms of phase transition and magnitude tuning (rather than the classical 'up or down' phenomenon) to identify only the treatment-responsive cells[43].

A broad impact of cocaine on transcription was observed across PFC, but with selective impacts on different cell populations at different stages (Supplementary Data 7). During the maintenance phase, the PFC exhibited an overall subtle change in transcription. About 10 or less genes were altered per excitatory neuron subtype (with no change in Exc-8, 10, 11) except Exc-6 where more than 40 genes changed (Fig. 6d). The affected neurons, including Exc-6, were mainly of L2/3 (Supplementary Fig. 9c).

However, prominent impact was observed upon drug withdrawal. Effect was smaller at 48 h, but became more pronounced at 15 day with all excitatory subtypes impacted to some degrees. At 48 h, all subtypes except Exc-11 showed gene expression changes. Exc-1, 8 and 13 were the most strongly induced, while Exc-6 maintained induction (Fig. 6d). At 15 day drug withdrawal,

the number of genes affected increased in every single subtype with Exc-2, 6, 7, and 8 showing the greatest effects. Curiously, the number of altered genes reduced between 48 h and 15-day withdrawal in Exc-1. Overall, excitatory neurons located in deeper cortical layers (L5, L6) were affected more during withdrawal, especially between 48 h and 15-day (Supplementary Fig. 9c).

The orders of engagement and degrees of response of each cell subtype likely predict their primary, secondary or tertiary role in behavioral adaptation in addiction. Furthermore, it likely indicates that neural regulation (transcriptional) during maintenance is largely operated by the basal ganglia, and only withdrawal induces greater cortical involvement. Notably, progressive engagement of the deeper PFC layers (L5-6) upon withdrawal might explain psychomotor aspects of drug withdrawal and would be consistent with symptoms seen in humans[44].

We also find that transcriptional changes induced by cocaine in PFC are cell type specific. A low overlap of commonly affected genes among the different cell subtypes indicates that expression of distinct genes is impacted in individual cell types (Fig. 6e). The effects are most prominent at the 15-day withdrawal time point (Fig. 6e).

A very subtle impact of this IVSA regimen can be observed in few inhibitory neuron subtypes (Supplementary Fig. 9d). However, these observations are limited by the low abundance of the inhibitory cell populations and are consequently difficult to interpret. A limited impact on non-neuronal populations was also observed, with the endothelial cells most affected, followed by oligodendrocytes. Interestingly, contrary to neurons, non-neurons were impacted most strongly at 48 h time point and subdued at 15-day in general (Supplementary Fig. 9d).

Since excitatory neurons exhibit the most significant changes during IVSA, particularly at 15-day of drug withdrawal, we performed pathway analysis to identify the cellular functions affected by the transcriptional changes (Fig. 6f). A greater impact was observed on some subtypes, such as Exc-2, 6, 7, 8, and 10, as indicated by more enriched pathways impacted and/or higher significance (Fig. 6f). Exc-6 and Exc-2 are of L2/3 while Exc-8, 7, and 10 are L5 (Fig. 2a, d). Key neuronal functions like axon guidance, gap junction signaling and vesicular trafficking (clathrin mediated endocytosis) are all affected in these five excitatory neuron clusters that can impact structural as well as electrical properties (through gap junction) of synapses. Additional pathways that can support plasticity are also affected in subtypes Exc-7 and 8 (example: ephrin, integrin, Rac, Gastrin/Cck and ultimately, even synaptic LTP), suggesting their greater involvement (Fig. 6f).

However, the most significantly affected pathways across all these excitatory populations were oxidative phosphorylation, mitochondrial function and sirtuin signaling (Fig. 6f). Oxidative phosphorylation is pivotal in fueling pre- and post-synaptic processing[45], whose alteration impacts mitochondrial function, which in turn affects the sirtuin pathway[46]. These results suggest

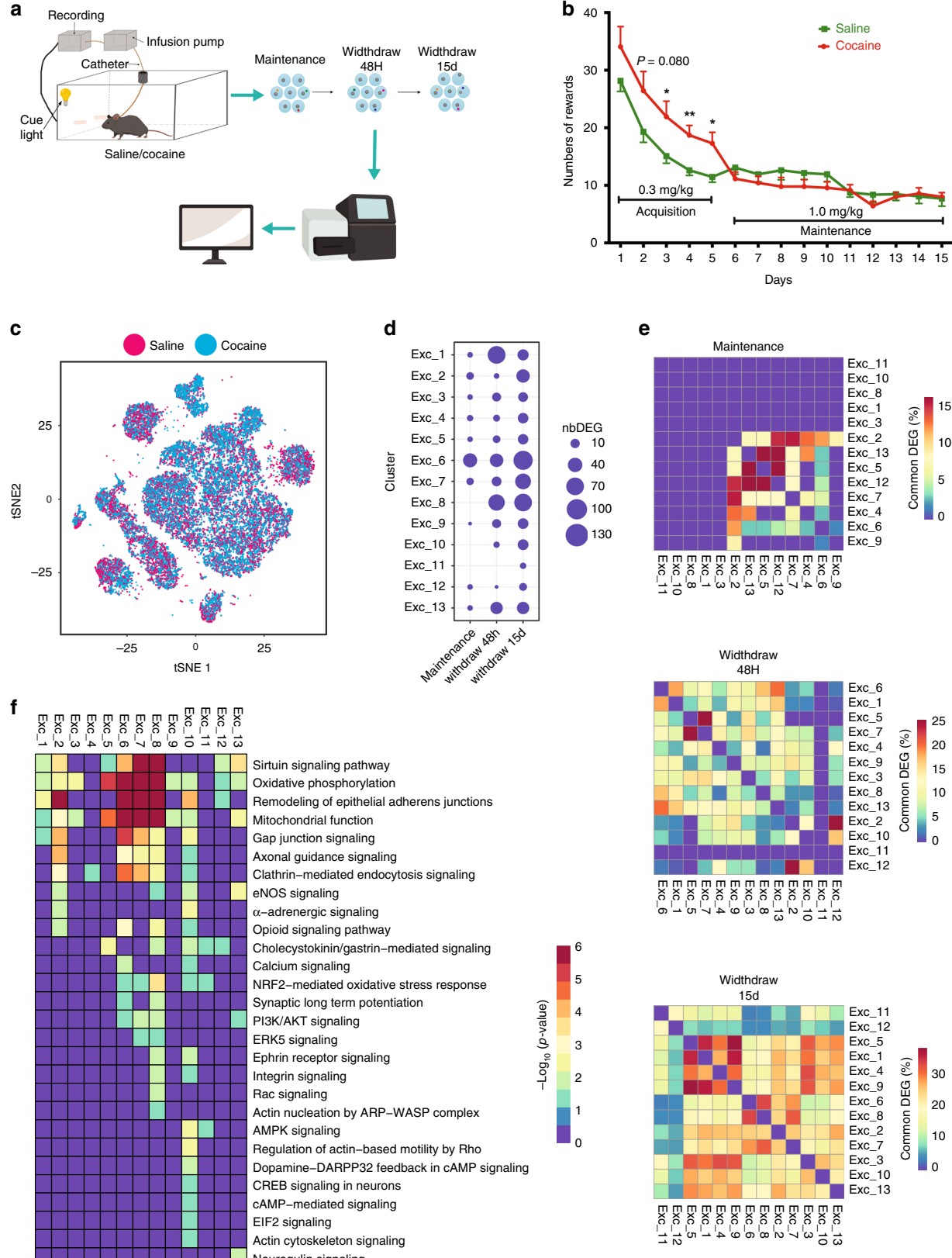

that neuroadaptations associated with addiction induces a strong metabolic and bio-energetic shift in the affected neuron populations. This is a fundamental difference from adaptive neuroplasticity of adolescence, where structural changes in neurons/synapses appear to play a more predominant role in neuroadaptation with altered oxidative phosphorylation, although affected, play a secondary role (Fig. 4e).

Interestingly, we found that the most strongly impacted subtypes in pathway analysis, Exc-5, 7, and 8, also express D1 dopamine receptor (*Drd1*) (Supplementary Fig. 9e), and therefore

**Fig. 6** Chronic cocaine IVSA induces transcriptional changes in multiple PFC cell types. **a** Schematic diagram showing the IVSA experimental setting and workflow. Created with BioRender.com. **b** Reward earnt by lever pressing in IVSA mice (saline and cocaine groups) through acquisition and maintenance of a 15-day period. **c** t-SNE-plot showing the uniform distribution of cocaine and saline samples in the different cell-types. **d** Dot-plot showing the number of differentially expressed genes between cocaine and saline in each excitatory neuron subtypes at each time point (cutoff: FC > 1.5 and SC2P model phase2 FDR < 0.05, empirical Bayes statistic, limma package). **e** Heatmap showing the**e** percent of commonly differentially expressed genes between the different clusters. Clusters tend to show cluster-specific differentially expressed genes with the highest overlap observed around 30% during the 15-day withdrawal. **f** Heatmap showing the -log$_{10}$(p-value) of the pathways enriched for differentially expressed genes in the different clusters at the 15-day withdrawal

can directly respond to dopamine. *Drd1*is also expressed by Exc-12 and 13 (Supplementary Fig. 9e) which are moderately impacted (enrich oxidative phosphorylation or mitochondrial function pathways). Thus, it is likely that one or more of these PFC neuron subtypes (5 out of 13 excitatory) respond to dopamine released from midbrain projections and participate in reward processing in conjunction with regions like the VTA and NAc. Collectively, our data suggest that the impact of cocaine in PFC is highly cell subtype-specific and temporally regulated. Our data also suggest that adaptive and induced neuroplasticity may involve distinct mechanisms of transcriptional adaptations across different cell types of the PFC.

**Data Visualization**. Data presented in this manuscript can be visualized and browsed at the following website: https://www.zhanglab.tch.harvard.edu/neuro-group/PFCExplorer

## Discussion
In this study, using scRNA-seq, we revealed the identity of the many discrete cell types in mouse PFC and defined their unique transcriptional features. We demonstrated how widespread transcriptional adaptations occur in each cell type in early post-natal life during spontaneous experience-dependent plasticity, as well as in later life under a compulsive cocaine addiction conditions. Comparative transcriptome analysis revealed critical insights into the functional implications of these transcriptional adaptations, potential epigenetic mechanisms underlying the changes, and how disruption of the process may contribute to major neuropsychiatric disorders.

The PFC displays rich cellular heterogeneity. As expected in cerebral cortex, we identified 8 broad cell types in PFC (including the excitatory neurons, inhibitory neurons and the different non-neuronal populations) (Fig. 1). The excitatory neurons could be broadly classified into 13 clusters, which could be further separated into 26 clusters with higher resolution. When compared with ALM and VISp, the PFC neuron subtypes exhibit highly distinct transcriptional properties. Compared to the excitatory neurons of VISp and ALM, the PFC has 5 (PFC9, 15, 16, 17) and 2 (PFC7 and 14) unique neuron subtypes respectively (Fig. 3c), which can be identified using distinctive markers (like *Nnat* or *Rspo2*) (Supplementary Fig. 4c, d). Even between the similar cell clusters in VISp and ALM, several hundred genes are still differentially expressed in PFC. Based on prior studies that demonstrated prominent transcriptional differences between the two geometric poles of the cortex (i.e., anterior and posterior)[47], a difference between PFC and VISp may be expected. However, the ALM, which is anatomically adjacent to the PFC, still exhibited marked differences in transcriptional features. Thus, in the cortex, in addition to the anatomic location, function and/or network connectivity appear to play an important role in deciding the transcriptional fate of an excitatory neuron.

For inhibitory neurons, which are far less abundant, broad clustering detected 12 distinct subtypes. Prior studies indicate that inhibitory neurons show uniform molecular features even between distant cortical areas[14]. In general, our results are

consistent with these observations. In addition to detecting the most abundant subtypes (eg. *Sst*, *Pvalb* or *Vip*)[14], we have detected and identified distinctive or combinatorial markers for at least four less abundant subtypes (Inhib-3, 7, 8, and 12) (Fig. 2g, Supplementary Figs. 3b, c). Further characterization of these distinct subtypes in future may reveal more PFC-specific functional (or pathological) features.

Psychiatric disorders are complex, poorly understood and mostly incurable[7]. Often, they are even difficult to definitively diagnose. Genome wide association studies (GWAS) have linked these disorders to a group of genes[41]. However, a causal relationship between a gene function and a disorder, as well as the underlying mechanism is difficult to establish. Furthermore, it is also unrealistic to perform transgenic animal-based studies on the large number of associated genes without building strong hypotheses. We demonstrate in this study that with the power of single cell analysis, the cellular basis/target of each GWAS mutation can be determined. Cell type- or neuron subtype-specific expression pattern of a disease-relevant gene can be determined by performing scRNA-seq of brain samples collected from animal models and postmortem human. A patient with a particular disorder can be diagnosed based on the specific disease-related mutation identified by sequencing of genomic DNA isolated from blood sample or oral swab. Thus, with knowledge of the cellular basis of pathogenesis, understanding of the disease mechanisms/manifestation, progress and targeted cell-type specific therapy can be developed/executed. This can be a remarkable step forward in heterogeneous tissues such as brain where distinct neighboring cell types can have discrete functions.

Our integrative analyses have allowed us to: (a) identify disease-relevant cell types for a disorder; (b) identify unique dysfunctions in individual cohorts (due to impact on very specific subtypes); (c) deduce a potential functional basis of a disorder (such as excitatory/inhibitory balance[48,49], lack of myelination[50] etc.) based on the cells affected. This will contribute not only to our understanding of disease mechanisms, but also design of more specific therapeutic strategy.

For some conditions, such as schizophrenia, bipolar disorder or autism, strong influence on excitatory or inhibitory neurons were observed; while for some other disorders (eg. bipolar or Alzheimer's), non-neuronal cells have equal or even greater contribution (Fig. 5d). It is now appreciated that many psychiatric disorders may emerge from dysfunctions of non-neuronal cells, which cause disruptions of major support systems like myelination or blood-brain barrier. More research in this area is essential in future for better mechanistic understanding.

Adolescence is known to be a major landmark of postnatal PFC maturation. Many psychiatric disorders associated with genetic defects surface during adolescence[10]. Yet which type of cells in PFC exhibit transcriptional dynamics during this period, and by what means, have remained unknown. Interestingly, we find that every PFC cell type undergoes substantial transcriptional changes during this period. While excitatory neurons show more wide-spread changes than inhibitory neurons, the lower abundance of the later underpowering the differential gene expression analysis cannot be fully omitted as a possibility. Although we could

generate 26 "high resolution" clusters for excitatory neurons, we performed gene expression analyses using the 13 "low resolution" clusters for a better power and higher accuracy to determine differential transcription.

Transcriptional changes affecting multiple common signaling pathways (like Rho, Rac, integrin, ephrin etc.) across most excitatory neuron subtypes during adolescence strongly implicate widespread structural plasticity (of axons, dendrites and synapses) during this stage (Fig. 4e). However, many cells also undergo modulation of specific pathways characteristic of their own subtype. For example, Exc-7 and 8 undergo modulation of dopamine receptor signaling (Fig. 4e). Indeed, both subtypes show *Drd1* expression and undergo strong transcriptional modulation in subsequent cocaine IVSA experiment (Supplementary Fig. 9e). These findings align with preexisting notion that exposure to addictive substances during adolescence can impair PFC maturation, and open doors for more precise mechanistic and therapeutic studies[51].

Similarly, based on transcriptional modulation of key GWAS candidates, target cell subtypes can be identified for mechanistic or therapeutic studies in various neuropsychiatric disorders. For example, Exc-13 records highest number of altered GWAS candidate genes in schizophrenia (Fig. 5e), and many genes like *Hs3tst5* (<−4 fold), *Kif5c* (<−2 fold) or *Wwox* (>1.5 fold) are regulated in this subtype during adolescence, suggesting a vulnerable neuron subtype in schizophrenia pathogenesis.

Less is known about cell type-specific regulation of the neuronal epigenome in later postnatal life. Interestingly, we find that the widespread gene regulation is accompanied by differential expression of several histone modifying enzymes or enzyme-complex members in PFC (Fig. 4f, Supplementary Fig. 6c). Each subtype regulates a combination of these (few common and many unique), with both activators and repressors. Therefore, it is plausible that cell-type specific combinations of locus-specific regulators modulate broad transcriptional changes in each subtype. Prior studies show that cortical DNA methylation takes place during postnatal 1–4 weeks[40]. Thus, while histone modifiers are clearly predominant regulators, a marginal role of DNA methylation in some of the initial impact of P21→P60 cannot be fully ruled out.

Although occurring much later in life, transcriptional changes from cocaine-induced plasticity is still significant and impacts multiple neuron subtypes. Interestingly, the impact is negligible during maintenance and dramatically increases during prolonged drug withdrawal. This implies that acquisition of reward possibly has a greater impact on basal forebrain structures of the reward circuit like the nucleus accumbens, and a craving response upon withdrawal greatly engages the PFC. Although never studied at single cell resolution, prior human studies have predicted such possibilities[52]. Importantly, our study provides evidence that PFC, unlike other brain regions, retains significant plasticity much later in life, and cell type-specific transcriptional responses can adapt to stimuli such as cocaine withdrawal.

At the maximum impact upon 15-day withdrawal, we detected strong response in some excitatory subtypes such as Exc-6, 7, 8, and 13 (Fig. 6d). Interestingly, some of them (Exc-7, 8, and 13) express dopamine receptor *Drd1* (Supplementary Fig. 9e) and may be directly responsive to dopamine released by VTA neurons projecting to PFC. The selective role of these subtypes is further supported by the fact that some other *Drd1* expressing subtypes (eg. Exc-5 or 12) do not show a strong response to cocaine. While further study is needed to demonstrate functional involvement of the specific subtypes, the findings support the existence of specific neuron subtypes in PFC that contribute to the process of drug addiction. Further characterization of these specific neuron populations in future will help not only in understanding the

mechanisms, but also in developing targeted therapies for drug addiction.

Notably, transcriptional changes associated with adolescence showed distinct functional differences relative to that induced by cocaine. While adolescence broadly invoked structural plasticity of axons and synapses, cocaine predominantly altered metabolic parameters to support synaptic potentiation (Figs. 4e, 6f). However, despite such differences, some common subtypes (like Exc-8 or 13) were strongly induced in both events. This may imply that there are fundamental differences of mechanisms/response underlying naturally adaptive and externally induced plasticity. However, the age difference between the two models (adolescence- P21 and cocaine starting at P60), which can naturally impact the basal adaptability/plasticity of the brain may also have some contribution towards the difference. That cocaine alters oxidative phosphorylation and other metabolic parameters in PFC has been observed in prior human studies[53], we reveal that this function is altered only in specific cell types. However, it should be noted that upregulation of pathways like sirtuin signaling, mitochondrial function and oxidative phosphorylation also contribute to neuronal stress. This can in turn cause excitotoxicity and deterioration of neural health and excitability that eventually resulting in a long-term deleterious effects of drug addiction.

Collectively, these cocaine-taking associated transcriptome changes provide supporting evidence that cocaine-taking does have an impact on the transcriptome of the adult PFC and that the impact could be variable across different neuron sub-types. Importantly, this paradigm demonstrates the degree of transcriptional adaptability that different neuron subtypes retain even in the adult PFC. However, it must be noted that there are many factors modulating drug self-administration[54], and further studies are needed for a complete understanding of neural mechanisms underlying the pathogenesis of voluntary drug taking.

While transcriptional changes are inevitable for the continuously adapting PFC neurons, it is essential to relate these adaptations to functional implications. Several genes altered in both paradigms (adolescence and cocaine-taking) bear strong testimony to functional implications and thus are worth to be explored further in the future. For example, *Marcksl1*, whose expression is dynamically regulated during adolescence (Supplementary Fig. 5a–c), is highly expressed in P21 when key functions involving this gene, such as cytoskeletal regulation, protein kinase C signaling or calmodulin signaling[55], are on the rise. While there are limited studies on the role of this protein in CNS neurons, strong functional implications emerge from this correlation. Similarly, *Ptgds* upregulated at P60 (Supplementary Fig. 5d), is believed to have neurotrophic function[56] and high frontal lobe expression[57], yet little understanding of its specific function, particularly within this time window, is available.

On the other hand, several altered genes in P21-P60 that were detected in GWAS for example, *Ddn*[58], *Slc17a6*[59] or *Penk*[60] imply direct association with major neuronal functions. Similarly, several genes implicated in cocaine withdrawal for example, *Nrgn*[61], *Cck*[62] or *Rab3a*[63] (which also change in more than one cluster) have widely known neuronal function. Knockout mouse models of these genes exhibit characteristic neurological functional deficits (*Rab3a*[63], *Penk*[64] or *Cck*[65]). Thus, our current study shed light on the potential cellular mechanisms of the neurological defects.

In summary, our study presents a comprehensive account of the widespread transcriptional dynamics in the postnatal PFC. It documents cell type-specific transcriptional adaptations across the PFC under two crucial conditions: adolescence and drug addiction. Results from this study reveal insights into the cell subtypes and molecular pathways that likely play an important role in a

broad range of cognitive and psychiatric disorders related to PFC, paving the way for not only advanced mechanistic understanding but also highly targeted therapeutic designs in the future.

## Methods

**Animals**. The animal use and experiments were conducted in compliance with the institutional IACUC Committee (of the HCCM). All male mice were used in the current study. For the P21 data collection, 4 mice were used. For the cocaine IVSA, 12 mice were used in each group (saline and cocaine). Each biological replicate was generated by pooling brain tissue from two mice.

**Tissue collection and library preparation**. Twelve independent biological replicates were performed, with brains from two mice used in each replicate. For single-cell dissociation, the mice were anesthetized with isoflurane and the brains were dissected and transferred into ice-cold Hibernate A/B27 medium (60 ml Hibernate A medium with 1 ml B27 and 0.15 ml Glutamax). The brains were sliced into 0.5 mm slices in ice-cold Hibernate A/B27 medium with brain matrix and the prefrontal cortex (PFC) was removed from each slice under dissection microscope. PFC tissues from two mice were pooled and dissociated into single-cell suspension. Briefly, the tissues were cut into small pieces and incubated in papain solution (Hibernate A-Ca medium with 2 mg/ml papain and 2X Glutamax) at 30 °C for 35 min with constant agitation. After washing with 5 ml Hibernate A/B27 medium, the tissues were triturated with fire polished glass Pasteur pipettes in 2 ml Hibernate A/B27 medium to release single cells, which was repeated for another two times. The 6 ml single-cell suspension was pooled and loaded on a 4-layer OptiPrep gradient and centrifuged at 800 g for 15 min at 4 °C to remove debris. The cells were then washed with 5 ml Hibernate A/B27 medium followed by 5 ml DPBS containing 0.01% BSA. The cells were spun down at 200 g for 3 min and re-suspended in DPBS containing 0.01% BSA. A 10 μl aliquot was stained with Trypan Blue and cell number was counted. For single-cell RNA-seq, the cells suspension was diluted to 300–330 cells/μl and captured with 10X Chromium platform (10X Genomics, CA). Reverse transcription, cDNA amplification and library preparation were performed according to the protocol from the manufacturer.

**Preprocessing of single-cell gene expression data**. Raw reads were pre-processed using the cellranger software (v.1.3.1)[66]. The "cellranger mkfastq" command was used to demultiplex the different samples and the "cellranger count" command was used to generate the gene-per-cell expression matrices for each sample by aligning the reads to the mm9 genome and quantifying expression of the Ensembl genes (Mus musculus NCBIM37 release 67). In total, 12 single-cell gene expression matrices corresponding to the 12 biological replicates were generated.

Single cell RNA-seq data was mainly analyzed by R package Seurat (v2.1.0)[17]. Briefly, all the expression matrices of the 12 samples were merged into a global Seurat object using the "MergeSeurat" function. The gene expression profile of each single cell was then normalized to counts per-million (cpm) and natural log transformed. As an initial quality control, cells that: have a potential mitochondria contamination (>10% of their total transcripts from mitochondrial transcriptome) or likely represent double-droplets (cells expressing more than the 99th percentile of the number of genes, 4766 genes) were removed. Abnormally high mitochondrial mRNA is an indication of dead or dying cells. Incorporation of these cells into the analysis can skew the results and introduce errors in detection of differentially expressed genes. Accordingly, it becomes a routine QC practice to filter out these cells in data analysis[67,68]. Since previous studies have showed that neuronal cells have a higher number of mRNAs (UMI) and genes expressed than that of non-neuronal cells[69,70], which was confirmed in our dataset (Supplementary Fig. 1b), we decided to filter cells using ≥800 genes/cell for non-neuronal cells, and ≥1,500 genes/cells for neuronal cells. We also removed all the mitochondria genes and ribosomal genes in the analysis. In total, an expression matrix including 20,718 genes and 27,702 cells was used. After clustering and manual assessment of double droplets a total of 24,822 cells were retained (11,886 saline and 12,936 cocaine).

**Broad clustering analysis and t-SNE plot generation**. To remove any potential batch effect between saline and cocaine samples, we run the CCA analysis[17]. Saline and cocaine cells were first grouped into different Seurat objects. For each object, genes showing a dispersion (variance/mean expression) larger than two standard deviation away from the expected dispersion were selected as variable genes using the Seurat function "FindVariableGenes". To run the CCA analysis, we selected the common variable genes between the top 2000 variable genes in saline and cocaine samples. Next, CCA analysis was performed by calling the functions "RunCCA" and "AlignSubspace". The aligned canonical correlation vectors showing a biweight midcorrelation ≥ 0.15 were selected to generate the t-SNE plots. For broad cell classification (Fig. 1b), we generated an initial set of 32 clusters using the first 12 aligned canonical correlation vectors by calling the "FindClusters" function. Clusters were then aggregated into the global families using the expression of known markers (Fig. 1d).

**Generation of the excitatory neuron clusters**. The excitatory neurons were classified using two levels of hierarchy. The first level captures the main canonical cell types and the second level captures more specific subpopulations. To generate the canonical cell types, we first generated an initial set of clusters using the "FindClusters" function with a resolution of 2. Then, we gradually merged similar clusters until no clusters could be merged using the "ValidateClusters" function. Briefly, a pair of two clusters were merged into one cluster if: (1) it showed a mean graph connectivity larger than 90% of all pairwise clusters connectivity in the constructed Shared Nearest Neighbor (SNN) graph; and (2) if the linear-SVM classifier has <90% accuracy in segregating between the two clusters using the top 100 genes from the first 5 PCs as features.

**Identification of cluster markers**. The cluster-specific markers were identified by detecting the differentially expressed genes between the given cluster and the other clusters. Specifically, the cluster markers were calculated by using a Bonferroni-corrected negative binomial generalized linear model (*FindAllMarkers* function of the Seurat package, test.use = "negbinom") while controlling for the number of UMI in each cell, the percent of mitochondrial genes, and sample-to-sample variation.

**Fluorescence in situ hybridization**. Mice were trans-cardially perfused with ice-cold PBS followed by ice-cold 4% PFA. Brains were harvested and incubated overnight in 4% PFA at 4 °C. Brains were cryopreserved in 30% sucrose solution in PBS at 4 °C. Sections were cut at 14–20 μm. Multi-channel fluorescence in situ hybridization was performed using RNA-Scope reagents and protocols (ACD Bioscience, CA) following manufacturer's instructions.

**Functional enrichment analysis**. The functional enrichment analysis was performed using IPA (QIAGEN Inc., https://www.qiagenbioinformatics.com/products/ingenuitypathway-analysis)[71]. Analysis was performed for each cluster with a cutoff of FC > 1.5 and q-value of 0.05 for the regulated genes. Analysis was conducted under "canonical pathways" and "disease and functions" that annotate all GO pathways in IPA. A right-tailed Fisher exact test p-value cutoff of at least 0.05 was applied in reporting any significantly enriched functional pathway. The associated GO and pathway enrichment plots were generated using the ggplot2 package.

**Generation of t-SNE plots and heatmaps**. The significant PCs (p-value < 1e-3, determined by JackStraw method in Seurat package) were used to generate the two-dimensional t-Distributed Stochastic Neighbor Embedding (2D t-SNE) which projects the data into a 2D-space. Heatmaps were generated using the R/Bioconductor package ComplexHeatmap[72] or the pheatmap R package. All the other plots were generated using the ggplot2 package (Wickham 2009). The circular heatmap in Fig. 5d was generated by calculating the percent of the cluster- and cell-type specific GWAS genes in a specific disease to the total number of GWAS genes to that disease and presented using the circlize package[73].

**Corresponding cell clusters between PFC, VISp, and ALM**. To detect similar cell sub-types between PFC and the other cortical regions, we compared PFC with VISp and ALM cell clusters independently. Since the two datasets were sequenced at different depths, conventional correlation analysis can lead to inconsistent results. We thus first used canonical correlation analysis (CCA)[17] to find a projection space that maximize the correlation between the two samples so that batch effect can be removed. We then built a similarity network between the cells using the first 20 aligned canonical correlation vectors. The MetaNeighbour method[74] was used to perform neighbor voting and the degree of similarity between PFC and VISp or ALM cell subtypes were calculated. If two clusters had a similarity degree ≥90% we considered them to be similar, and they are clustered together (Fig. 3c). Clusters that showed <90% similarity to any cell-type from the other cell population were considered as unique (highlighted in bold and marked by * in Fig. 3c).

**Detecting differential gene expression PFC vs. VISp or ALM**. As the canonical correlation analysis (CCA) can only present cells into another projection space that brings similar cells into near proximity, it does not correct the gene expression between two datasets. Hence, we tried to find non-batch related differentially expressed genes between similar cell-types in PFC and the other brain regions. Each group in Fig. 3c was processed separately. For each group, PFC cells were merged together and the corresponding cells from the other tissue were merged together. Then, the 'multiBatchNorm' function of the scran R/Bioconductor package[75] was used to rescale the datasets to similar sequencing depth. Then, differential gene expression analysis was performed using a likelihood-ration test[76] (Seurat function "FindMarkers" with parameter test.use = "bimod").

**Comparison between PFC cells of P21 and P60 mice**. Before assessing the correspondence between P21 and P60 cells, we first filtered the P21 data set. After removing low quality cells (<800 detected genes and cells having >10% mitochondria transcriptome) and double-droplet cells, a total of 10,646 high quality cells were obtained from P21 PFC.

A CCA analysis was performed to align the P21 on top of the P60 cells. To assign cell identity to the P21 cells, a bootstrapped correlation analysis was performed. Briefly, the top 100 genes with the highest CC score from each of the first 20 CC were selected and merged together to get a list of 943 genes. Then, 100 bootstrapped correlations were run. In each iteration, we randomly selected 80% of the 943 genes and calculated spearman correlation (to limit the bias induced by different sequencing technologies). Next, each P21 cell was assigned to a cluster in which it showed >50% similarity probability. Only 9.1% of the P21 cells did not show a high similarity to the PFC cells. When we checked their distribution in the t-SNE they showed a random localization in the t-SNE plot indicating that they could be faithfully classified due to technical issues rather than representing new cell-types. The differentially expressed genes between P21 and P60 cells for each cluster was performed using the "FindMarkers" function from the Seurat package using a likelihood ratio test and correcting for the number of detected UMI bias.

**Association of GWAS genes to cell clusters**. To identify the GWAS candidates that show a specific enrichment pattern in the different cell subtypes, we first downloaded a list of GWAS candidate genes and their associated disease from the NHGRI-EBI GWAS catalog (version 1.0.2)[77]. We only considered the GWAS genes that have an exonic mutation and which have been detected by published studies that pass the NHGRI-EBI GWAS catalog eligibility criteria To detected GWAS genes that show a specific enrichment pattern, we used the following two criteria: (i) the genes should show a non-uniform expression along all the clusters (Shanon-entropy based test, RNentropy package[78], $p$-value < 1e-5), ii) the cells expressing the gene should be concentrated in subset of clusters and not scattered along all the clusters in a uniform manner Shanon-entropy based test, RNentropy package, $p$-value < 1e-8) The second criteria enables us to detect potential false-positive genes that broadly expressed by showing a variability in their expression along the cell-types. To annotate the enriched (1) or non-enriched (0) clusters for each of the non-uniformly expressed genes, we ranked the clusters by their increasing expression and the clusters having an average expression larger than knees value in the plot were selected (kneepointDetection method, SamSPECTRAL package[79]). To avoid outlier effect, the mean expression of each cluster was calculated using cells with an expression value <99th percentile for each gene.

**Detection of genes after cocaine self-administration**. To detect the genes affected by cocaine in each cluster, we used a "Two-phase transcription" model (SC2P package[43]), which assumes that cells in a homogeneous population response non-homogeneously to a stimulus. Thus, for each gene, cells that represent technical dropouts or lowly expressing cells that might represent random initialization are classified as Phase-I cells, and cells in which the gene is 'on' are considered Phase-II cells. Differential gene-expression analysis is focused on the marginal changes in the Phase-II cells as they represent changes induces by cocaine stimulation.

**Protocol for cocaine intravenous self-administration (IVSA)**. The self-administration procedures consisted of four procedures, food training, catheterization, re-baseline food training, and cocaine administration.

One day prior to the first food training session, mice were food restricted to about 85% of their baseline body weight. Then the mice were subjected to a food training for about 1 week, 1-h per day. The food training session allowed the mice to associate the active level pressing with food reinforcers, i.e., 20-mg food pellets (Bio-Serv, NJ). Mice were trained in a progressively increased fixed-ratio (FR) from FR = 1 to FR = 5, time out (TO) 20 s.

Once stable level responding was achieved (>30 pellets per session), mice were subjected to catheterization. Prior to the catheterization, mice were returned to normal food chow and water *ad libitum*. Under 1.5% isoflurane and oxygen anesthesia, the right jugular vein was catheterized. Implanted catheters were flushed daily with saline solution that contained pre-diluted heparin.

Following catheterization, mice were allowed recovery for >2 days, and subjected to re-baseline food training (FR = 5, TO = 20 s) again to achieve a stable food response level (>30 food pellets per session). Once stable food response was achieved, mice were subjected to cocaine self-administration. During cocaine self-administration, a 1-h daily administration of cocaine solution was allowed. During the initial IVSA acquisition session, which lasted for 5 days, 0.3 mg/kg cocaine was administrated through an automatic pump delivery to the jugular vein (0.03 ml infusion volume per infusion). Only active lever pressing can result in successful drug delivery. Successful drug response was deemed as at least 6 infusions per session. After the initial acquisition session, the mice were subjected to a higher dosage of cocaine (1.0 mg/kg) for another 10 days.

Mice that experienced the same IVSA procedure serve as a control except that cocaine was replaced with saline solution.

## Data availability

The raw data and the count matrices are available under accession number GEO: GSE124952.

## Code availability

All the codes used in this study are available upon reasonable request.

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

## Acknowledgements

We thank Dr. Zhiyuan Chen for help with mouse colony and sequencing, along with Mr. Qianzong Yin. We thank the support of the HMS Neurobiology Imaging Facility (supported by NINDS P30 Core Center Grant #NS072030) and its staff Ryan Carelli and Michelle Ocana. This project was partly supported by NIDA 1R01DA042283, Open Philanthropy Project LLC ("Open Phil LLC") grant support (#2018-191651) and HHMI. Y.Z. is an investigator of the Howard Hughes Medical Institute.

## Author contributions

Y.Z. conceived the project; A.B., R.C., W.C. and L.M.T. performed the experiments; M.N.D. analyzed the high-throughput dataset; A.B., M.N.D. and Y.Z. wrote the manuscript.

## Additional information

**Competing interests:** The authors declare no competing interests.

