## [Peer Review File · Nature Communications]

Reviewers' comments:

Reviewer #1 (Remarks to the Author):

In this manuscript titled "Cell type-specific transcriptional programs in mouse prefrontal cortex during adolescence and addiction", the authors comprehensively analyzed the transcriptome of single cells within PFC as well as the dynamics during adolescent development and drug addiction. The single-cell resolution transcriptomic analyses reveal cell type specific changes in adolescence and drug addiction, which reveal the cellular and transcriptional basis for these processes. Previous genome-wide analysis mainly use the whole dissected parts of the brain tissue as the experimental materials which are complex mixes of neuronal and non-neuronal cells, leading to diluted meaningful transcriptional changes and incapability of telling the cellular basis of the changes. Therefore, this study is a major advance of understanding the transcriptional outcome of development and drug addiction which provides a rich resource for further mechanistic study in the future and offers a new angle to understand the brain in physiological and neuropsychiatric disease states at the highest resolution. I believe this will be of great interest to the community. Here are several comments to improve the manuscript.

1. The differential expression analysis between P21 and P60 used a cutoff of 0.05 FDR with 50% change which was a relatively loose cutoff. As the analysis was performed with a pure cell type where a more uniform change in expression is expected compared to a chunk of tissues. Will the author explain in the text why this cutoff is chosen for the analysis here and following figures? How about using a more stringent cutoff to focus on the mostly changed genes?
2. The Fig. 5e is hard to understand at the present form of presentation. You cannot tell Ex-13 or Ex-8 described in the text easily in the figure. In addition, the color and number is not described in the figure legend.
3. The authors imply that the cell type specific gene expression for different disease associated genes can be used for diagnosis. Practically, it is not possible to take a slice of the brain from the patients for single-cell transcriptome analysis. How can the diagnosis be done? It is better to discuss the way to use this information for diagnosis in discussion or remove the claim for usage in disease diagnosis in the manuscript.
4. The scale bars in Fig. 6d are incorrect. For example, in the left one, it shall be 15 instead of 0.15.
5. Ex-7,8,13 express *Drd1* and show the strongest transcriptional response after cocaine withdraw. This phenomenon is very interesting, suggesting direct response to dopamine secreted from VTA neurons. It will be interesting to see more evidence for this although not required.

Reviewer #2 (Remarks to the Author):

The manuscript presented by Bhattacharjee et al. is a transcriptional characterization of prefrontal cortex (PFC) neurons to understand the diverse cellular populations that comprise this brain area as well as how they diverge throughout development and in response to behavior. Specifically, using single-cell RNA seq, the authors identify various neuronal subpopulations within the traditionally used excitatory and inhibitory subclasses – highlighting that our genetically defined cellular identifiers are likely not selective enough. Further, using FISH these marks are validated and show genetic markers that can be used to differentiate unique subpopulations in the PFC. Further, the authors characterize how PFC neurons change between the adolescent and adulthood and the effects of cocaine history on adult PFC gene expression. While this project lacks a clear hypothesis it could be of broad interest to researchers focused on the cortex or on genetically defined cellular populations. However, while the data is incredibly powerful and the techniques elegant, there are concerns with the analysis that temper enthusiasm for the manuscript in its current form. My concerns are listed below as major and minor:

Major:

1. The biggest concern is in terms of statistical validity. The authors provide in depth analysis of genetic differences that occur throughout development (P21 vs P60) in the prefrontal cortex. However, a major confound of cell clustering is that the total number of cells analyzed directly affects the unique cell clusters identified. As reported in the methods, only 10,646 cells were analyzed for P21 PFC analysis, while 27,702 cells were analyzed for P60 PFC. Total cells analyzed for P21 need to be comparable to those of P60 in order to reliably compare the clusters subsequently generated. Similarly, the number of cells analyzed in cocaine self-administration should be reported with this in mind.

At the very least, this should be validated with permuted subsampling from the larger dataset to see if the effects are reproducible with a smaller sample size.

2. There are a large number of concerns with the self-administration figure including the fact that no behavioral data is shown and a number of issues with time from self-administration and the activity-dependent gene lists that are shown which would select for high levels of correlation across all neuron subtypes in groups that are in closer proximity to the self-administration experience. In order to control for drug effects vs non-specific effects of behavior experience (saline groups do not control for this) a large number of additional experiments would have to be run. However, the manuscript contains a large amount of data as is and the cocaine self-administration data seems like an add on that is outside of the scope of the current work. I would suggest removing these data all together and publishing them separately.

Minor:

1. The tissue used in this sequencing combines both prelimbic (PrL) and infralimbic (IL) tissue (which is understandable given limitations/demands on mouse tissue). However, these regions have distinct functional roles throughout development and in the adult (in particular acquisition and maintenance of cocaine self-administration). It would be relevant to the field to see if in their data sets, do PrL and IL populations differ from each other throughout development and throughout cocaine history. While it is difficult parsing these populations, there are a subset of genes that are differentially expressed that could allow transcriptome-based clustering (such as *Sdk2*, and the *Fgf* family of genes as described by Cholfin and Rubenstein, 2007).

2. There are inconsistencies in the data presentation that should be corrected for clarity. For example, heatmaps in Fig. 3C should provide both X and Y axes for the cluster by cluster comparisons.

Reviewer #3 (Remarks to the Author):

The manuscript by Aritra Bhattacharjee et al. harnesses the power of scRNAseq to deconvolute cell types in the PFC. They then perform a useful comparison to previously identified cell types in ALM and VISp. They also relate these cell types/genes to GWAS studies for several neuropsychiatric disorders. Having determined the cellular landscape of the PFC, they then proceed to examine changes in this region during postnatal maturation and during cocaine withdrawal. They find that the major difference in mRNA expression between P21 and P60 is largely genes/pathways related to cell structure, congruent with the idea that adolescence is a time of generating new synapses and pruning old ones. On the other hand, the main mRNA expression difference upon cocaine withdrawal is largely related to metabolic pathways, suggesting that the plasticity is not to do with neuron structural changes, but rather, metabolic changes. Overall, this is a well written and displayed manuscript, which makes a significant contribution to key areas of neuroscience.

Major point

1. Secondary validation: For differential gene expression in adolescence vs adulthood, or in the cocaine paradigm, no secondary validation is provided. At least some corroboration of the findings

using alternative methods should be provided. In addition, ideally, some functional corroboration of the gene expression changes would greatly strengthen this paper.

Minor points

2. CHD8 is associated with autism so it is interesting to find it upregulated in excitatory neurons. This finding could be further elaborated, for example, by analyzing/postulating downstream targets.
3. Sex and number of animals used in all studies should be mentioned.
4. For P21: semantically: Pg 3: did not, in fact, analyze "through adolescence", they compared P21 and P60. Especially since P21 is weaning day, these mice have not had "adolescent experience" of new independence as described on pg 10.
5. The authors should better explain the rationale for elimination of high-mitochondrial-mRNA-load cells.
6. Pg 6 paragraph 2: there is only one staining experiment shown, but is referred to as an example. If there were other staining experiments utilizing other markers, they should be shown in supplementary figures.
7. Cartpt is a diagnostic marker for some interneurons. Yet, only excitatory neurons were analyzed in the cocaine experiments with respect to Cartpt; this seems like an interesting piece of information that is missing.
8. Pg 9, Comparison of V1Sp and G11.....data is not shown. This should be added to supplementary figures ?
9. Fig. 4f, Please clarify the comparison being made.
10. ASD and ADHD are two very separate clinical diagnoses- They should not be lumped.
11. GO analysis: The authors should clarify methods for GO, add GO number to each graph/function shown. Which GO analysis was run? (Cell component, Biological process, or Molecular function) What is the p-value? Was there any correction of data or p-value? Which statistical parameters were used? Was a software used, or the webpage? When citing pathways that were most significantly changed, what was the p-value ?

Response to reviewers' comments

We sincerely appreciate the encouraging and constructive comments from all three Reviewers. We address their comments point-by-point below:

Referee #1 (Remarks to the Author):

In this manuscript titled "Cell type-specific transcriptional programs in mouse prefrontal cortex during adolescence and addiction", the authors comprehensively analyzed the transcriptome of single cells within PFC as well as the dynamics during adolescent development and drug addiction. The single-cell resolution transcriptomic analyses reveal cell type specific changes in adolescence and drug addiction, which reveal the cellular and transcriptional basis for these processes. Previous genome-wide analysis mainly use the whole dissected parts of the brain tissue as the experimental materials which are complex mixes of neuronal and non-neuronal cells, leading to diluted meaningful transcriptional changes and incapability of telling the cellular basis of the changes. Therefore, this study is a major advance of understanding the transcriptional outcome of development and drug addiction which provides a rich resource for further mechanistic study in the future and offers a new angle to understand the brain in physiological and neuropsychiatric disease states at the highest resolution. I believe this will be of great interest to the community. Here are several comments to improve the manuscript.

Response: We thank the reviewer for appreciating our study. We address the specific comments below:

1. The differential expression analysis between P21 and P60 used a cutoff of 0.05 FDR with 50% change which was a relatively loose cutoff. As the analysis was performed with a pure cell type where a more uniform change in expression is expected compared to a chunk of tissues. Will the author explain in the text why this cutoff is chosen for the analysis here and following figures? How about using a more stringent cutoff to focus on the mostly changed genes?

Response: The reviewer is correct that pure cell types would yield uniform detection of gene expression changes. However, this also means that even much smaller changes can be now detected accurately. In non-dividing cells such as neurons, transcriptional changes occur over a smaller dynamic range, and transcripts changing as much as half of the baseline ($\pm 50\%$ change) is believed to have significant functional consequences. Accordingly, ± 1.5 fold change has been set as cutoff in many recent landmark scRNA-seq studies in brain ^{1,2}. We therefore adhered to this paradigm. Also consistently, the fold-change difference distribution between P21 and P60 in our data shows that the majority of the changes are between 1- 4 folds (**Fig. R1a**). Increasing the FDR stringency to 0.01 (keeping the 50% FC cutoff) had negligible impact on the number of differentially expressed genes detected in the different clusters (**Fig R1b**).

Nevertheless, following the reviewer's suggestion, we used FC=2 cutoff and repeated the analyses, which identified the same cluster-specific signaling pathways. However, this new analysis significantly reduced the prediction power (**Fig. R1c**). Based on this result combined with the fact that recent similar scRNA-seq studies in brain used $|FC| > 1.5$ as a cutoff, we hope that the reviewer would agree that the currently cutoff parameters (0.05 FDR and $|FC| > 1.5$) provide a high sensitivity for interpreting the cluster-specific gene expression changes.

a

b

c

Fig. R1. The impact of different cutoff on the analysis of differential gene expression. a) Density plot showing the distribution of the fold-change values between P21 and P60. Negative values in the x-axis indicate down-regulation at P60, positive values indicate up-regulation at P60. **b)** Barplot showing the number of differentially expressed genes detected using a cutoff of $FC > 1.5$ and $FDR < 0.05$ (in red) and $FDR < 0.01$ (blue). **c)** Heatmap showing the decrease in the p-values of the terms detected using $FC > 1.5$ and $FDR < 0.05$.

2. The Fig. 5e is hard to understand at the present form of presentation. You cannot tell Ex-13 or Ex-8 described in the text easily in the figure. In addition, the color and number is not described in the figure legend.

Response: We thank the reviewer for pointing this out. We have modified Fig. 5e with discrete and clearly distinguishable colors. We have also included the description in the legend and emphasized in the text that Fig. S8a also describes and enumerates each cluster in an enlarged presentation for quick reference while reading the tSNE in 5e.

3. The authors imply that the cell type specific gene expression for different disease associated genes can be used for diagnosis. Practically, it is not possible to take a slice of the brain from the patients for single-cell transcriptome analysis. How can the diagnosis be done? It is better to discuss the way to use this information for diagnosis in discussion or remove the claim for usage in disease diagnosis in the manuscript.

Response: We actually imply that with single cell transcriptional profiles of the various PFC neuron subtypes, we can now identify the cell cluster that is likely most affected by a specific neuropsychiatric disease-associated gene mutation. This is based on our observation that many neuropsychiatric GWAS candidates are highly enriched in selective PFC cell clusters. Such information can greatly help prognosis, give valuable insight about disease mechanism/manifestation, and promote highly selective therapeutic targeting in future. For example, the specific mutation present in a particular patient can be identified by sequencing of DNA sample derived from patient blood/skin/oral swab etc. The mutation information can be associated with a specific cell type or neuron subtype based on our study, thus linking the patient to a likely cell type or neuron subtype defect. We have now clarified this point specifically in the Discussion (**Page 20**).

4. The scale bars in Fig. 6d are incorrect. For example, in the left one, it shall be 15 instead of 0.15.

Response: We thank the reviewer for pointing this out and we have fixed the error and changed the scale from decimals to percentage.

5. Ex-7,8,13 express *Drd1* and show the strongest transcriptional response after cocaine withdraw. This phenomenon is very interesting, suggesting direct response to dopamine secreted from VTA neurons. It will be interesting to see more evidence for this although not required.

Response: We agree with the reviewer that this is the next logical and exciting question. It is highly imperative that dopamine released into PFC will activate these cell populations. However, several parameters including degree or temporal order of induction of these cell types, anatomical subregions of dopamine release, or reward conditioning etc. must be accounted for in accurate analysis and interpretation. Thus, we plan to perform an independent follow up study in the coming years.

Referee #2 (Remarks to the Author):

The manuscript presented by Bhattacharjee et al. is a transcriptional characterization of prefrontal cortex (PFC) neurons to understand the diverse cellular populations that comprise this brain area as well as how they diverge throughout development and in response to behavior. Specifically, using single-cell RNA seq, the authors identify various neuronal subpopulations within the traditionally used excitatory and inhibitory subclasses – highlighting that our genetically defined cellular identifiers are likely not selective enough. Further, using FISH these marks are validated and show genetic markers that can be used to differentiate unique subpopulations in the PFC. Further, the authors characterize how PFC neurons change between the adolescent and adulthood and the effects of cocaine history on adult PFC gene expression. While this project lacks a clear hypothesis it could be of broad interest to researchers focused on the cortex or on genetically defined cellular populations. However, while the data is incredibly powerful and the techniques elegant, there are concerns with the analysis that temper enthusiasm for the manuscript in its current form. My concerns are listed below as major and minor:

Response: We thank the reviewer for appreciation of our work. We address the specific comments below:

Major concerns:

1. The biggest concern is in terms of statistical validity. The authors provide in depth analysis of genetic differences that occur throughout development (P21 vs P60) in the prefrontal cortex. However, a major confound of cell clustering is that the total number of cells analyzed directly affects the unique cell clusters identified. As reported in the methods, only 10,646 cells were analyzed for P21 PFC analysis, while 27,702 cells were analyzed for P60 PFC. Total cells analyzed for P21 need to be comparable to those of P60 in order to reliably compare the clusters subsequently generated. Similarly, the number of cells analyzed in cocaine self-administration should be reported with this in mind.

At the very least, this should be validated with permuted subsampling from the larger dataset to see if the effects are reproducible with a smaller sample size.

Response: We are sorry for not making it clearer. We have stated more details in methods now. For the comparison with P21 cells, we used only the 11,886 control P60 cells, so the number in both groups is in fact quite comparable. The 27,702 is the total cell numbers sequenced from the control and cocaine groups in P60. Also, to avoid the issue of dataset imbalance indicated by the reviewer, we assigned the cluster identity of the P21 cells using a bootstrapped correlation in which we selected the top variable genes (943 genes) and did 100 bootstrapped correlations. In each iteration, we randomly selected 80% of the variable genes and assigned the P21 cells which are most similar to the P60 cluster according to their spearman correlation (to limit the effect of sequencing depth bias). Each P21 cell was then assigned to the P60 cluster to which it had >50% chance to be assigned to. About 9% of the P21 did not show any similarity to P60 cells. However, these cells were uniformly distributed across the t-SNE plot (rather than forming any distinct cluster) (**Fig R2**): a clear hallmark of technical dropouts (low capture rate) rather than any biological variation. This further reinforces the clear match/comparability of the P21 and P60 samples.

For differential gene expression analysis, we used likelihood ratio test based on the negative binomial distribution in which we regressed-out the effect of sequencing depth by controlling for the number of detected UMI in each cell.

Fig. R2. t-SNE plot showing the predicted identity of the P21 excitatory cells. a) t-SNE plot showing the predicted identity of the P60 excitatory neurons to the adult 13 clusters. **b)** t-SNE plot showing the predicted identity of the P60 excitatory neurons to the adult 26 excitatory clusters. The P21 neurons showed similarity to the different clusters at different clustering resolutions, while the P21 cells that didn't have similarity were randomly distributed in the plots indicating that they don't share similar transcriptional profiles

2. There are a large number of concerns with the self-administration figure including the fact that no behavioral data is shown and a number of issues with time from self-administration and the activity-dependent gene lists that are shown which would select for high levels of correlation

across all neuron subtypes in groups that are in closer proximity to the self-administration experience. In order to control for drug effects vs non-specific effects of behavior experience (saline groups do not control for this) a large number of additional experiments would have to be run. However, the manuscript contains a large amount of data as is and the cocaine self-administration data seems like an add on that is outside of the scope of the current work. I would suggest removing these data all together and publishing them separately.

Response: We understand the reviewer's concerns and appreciate bringing these important issues to discussion. We explain below the original motivation of the study and the changes we have made to address this reviewer's concerns.

Indeed, we had presented the data for reward-earning by the IVSA mice as supplemental information in prior version of the manuscript. However, we acknowledge the emphasis laid by this reviewer on the importance of behavioral observations for this study and thus have now moved this data to the main figure (**Fig. 6b**).

The major motivation of the cocaine studies was to demonstrate that the transcriptional adaptability is retained in adult PFC neurons, which include their salient signaling features and cluster-specific response to cocaine-taking. Accordingly, we adhered closely to well-established paradigms and timelines conventionally used for cocaine IVSA, for example *Cameron et al* 2019³. We indeed achieve the goal by demonstrating a temporally variable, and a cell type-selective responsiveness in transcriptional plasticity across PFC. Figure 6e actually shows a low correlation (only up to 0.3 at most) amongst genes altered across the different cell types, implying largely unique transcriptional impact in each cell cluster. Therefore, as a model system, cocaine self-administration provides a proof-of-concept for cell type-specific transcriptional adaptations, which has been enthusiastically received by the other two reviewers.

We agree with the reviewer that there are many important questions that remain to be addressed to fully understand the mechanism of drug self-administration and that further experiments aimed at a complete understanding of the mechanism of drug self-administration is a long-term goal off the scope of the current manuscript. Although our IVSA cannot fully address specific transcriptional changes caused by the different factors (such as drug effect, behavior changes), it provides insights into cell type-specific transcriptional adaptations of a highly relevant region with a contingent addiction model. While further functional

characterization is necessary, the information derived here is useful and can guide functional studies in the future.

Therefore, we try to restrict our interpretation (and discussion) strictly within the limits of the acquired data. Considering the reviewer's concerns, we have now included additional discussions (**Page 23**) to restrict our statement. If the reviewer is satisfied, we urge to keep this dataset to illustrate the importance of phenotypic and functional segregation of adult PFC neurons, particularly given the other reviewers appear to be enthusiastic about this part of our story.

Minor concerns:

1. The tissue used in this sequencing combines both prelimbic (PrL) and infralimbic (IL) tissue (which is understandable given limitations/demands on mouse tissue). However, these regions have distinct functional roles throughout development and in the adult (in particular acquisition and maintenance of cocaine self-administration). It would be relevant to the field to see if in their data sets, do PrL and IL populations differ from each other throughout development and throughout cocaine history. While it is difficult parsing these populations, there is a subset of genes that are differentially expressed that could allow transcriptome-based clustering (such as *Sdk2*, and the *Fgf* family of genes as described by Cholfin and Rubenstein, 2007).

Response: The reviewer has raised a very interesting point here. We enthusiastically followed up based on available information. While cells could be segregated and represented as separate tSNE plots for the two different subregions, the distinction between the same cell types in the two different regions reflected only differential expression of very few genes (2-6 on average), raising the possibility of merely different transcriptional states as opposed to distinct cell types. Besides, there is also a difference in the abundance of some cell types between these two regions that can skew the analysis originating from a single data set (e.g. Ex-5, see below). Thus, a faithful prediction of distinct cellular phenotypes between PrL and IL based on existing transcriptomic data remains challenging. The ultimate tool for discrimination of cell types from PrL and IL would still be projection-specific neural tracing combined with transcriptome map/profile or specific cluster marker staining. We provide the details of our analysis below.

While Cholfin and Rubenstein provided a seminal account of differential genes between the PFC regions, their study was performed during early development. We found that many of these genes showed significantly lower expression later on in P21 or adult (**Fig. R3**). To investigate further, we searched through the literature and identified four potential markers that show high enrichment in the PrL tissue (*Dkk11* and *Ngb* for the superficial layers and *Fam84b* and *Fezf2* for the deep layers: Murugan M., et al. Cell, 2017)³ (**Fig. R4a and b**). Using these PrL-enriched genes, we segregated the excitatory clusters into PrL and IL cells depending on their enrichment/depletion for these genes (**Fig. R4c**). Briefly, for each cell we calculated the average expression level for these marker genes by subtracting the aggregated expression of randomly sampled control gene sets, cells with positive enrichment were classified as PrL and cells with negative enrichment were classified as IL. However, when checking the number of differentially expressed genes between PrL and IL, we found very marginal differences (2-6 for each cluster; except Ex-5 of L6 which is skewed by the under-representation of the cluster itself in IL). Accordingly, it left us with little confidence for the prediction and performing downstream analysis of this segregation.

Fig. R3 Expression of the genes identified by Cholfin and Rubenstein in excitatory neurons.
t-SNE plot showing the expression of the patterning genes detected by Cholfin & Rubenstein.

Fig. R4 Example of some PrL enriched marker. **a)** Allen Brain Atlas ISH images showing the expression of some prelimbic markers in superficial layers (*Dkk1*, *Ngb*) and deep layers (*Fam84b*, *Fezf2*), adapted from Murugan et al⁴. **b)** t-SNE plot showing the expression of the identified markers in our excitatory neuron dataset. **c)** t-SNE plots showing the segregation of excitatory neurons according to their enrichment for *Dkk1*, *Ngb*, *Fam84b*, *Fezf2*, *Sccpdh* into PrL cells (left panel, green color) and IL cells (right panel, purple color). **d)** Barplot showing the number of differential genes between the PrL and IL in each cluster.

In conclusion, while we appreciate the exciting idea proposed by this reviewer, it seems difficult to confidently prove the idea based on the current transcriptome analysis alone.

2. There are inconsistencies in the data presentation that should be corrected for clarity. For example, heatmaps in Fig. 3C should provide both X and Y axes for the cluster by cluster comparisons.

Response: We thank the reviewer for pointing this out. We have updated Fig.3C and showed both the X- and Y-axis. As the VISp and ALM cluster name (given by the initial authors) are very long, we shortened them and added a table showing the correspondence between the shortened names and the original cluster names in **Supplementary Table 3**.

Referee #3 (Remarks to the Author):

The manuscript by Aritra Bhattacharjee et al. harnesses the power of scRNAseq to deconvolute cell types in the PFC. They then perform a useful comparison to previously identified cell types in ALM and VISp. They also relate these cell types/genes to GWAS studies for several neuropsychiatric disorders. Having determined the cellular landscape of the PFC, they then proceed to examine changes in this region during postnatal maturation and during cocaine withdrawal. They find that the major difference in mRNA expression between P21 and P60 is largely genes/pathways related to cell structure, congruent with the idea that adolescence is a time of generating new synapses and pruning old ones. On the other hand, the main mRNA expression difference upon cocaine withdrawal is largely related to metabolic pathways, suggesting that the plasticity is not to do with neuron structural changes, but rather, metabolic changes. Overall, this is a well-written and displayed manuscript, which makes a significant contribution to key areas of neuroscience.

Response: We thank the reviewer for appreciating this work. We address specific comments below:

Major points:

1. Secondary validation: For differential gene expression in adolescence vs adulthood, or in the cocaine paradigm, no secondary validation is provided. At least some corroboration of the findings using alternative methods should be provided. In addition, ideally, some functional corroboration of the gene expression changes would greatly strengthen this paper.

Response: We agree that secondary validation is important to ensure proper working of the sequencing technique and analysis. We tested the P21 vs P60 comparison to validate. We tested multiple genes exhibiting dynamic changes between P21 and P60 that were identified in scRNAseq by smFISH. The results (**Fig. S5**) validated the findings of the scRNAseq.

We agree that functional corroboration can ideally strengthen the manuscript and raise its impact. However, we believe that the reviewer would agree that with description of so many clusters, their transcriptional adaptation and relevance in disease and development, additional data might exceed the scope of the current manuscript. However, there is ample evidence of specific functional implications within the described transcriptional changes. Therefore, in light of the reviewer's suggestion, we added a section (**page 24**) to discuss the functional implications of several key genes in the revised manuscript.

Minor points:

2. CHD8 is associated with autism so it is interesting to find it upregulated in excitatory neurons. This finding could be further elaborated, for example, by analyzing/postulating downstream targets.

Response: It is true that autism is one of the least understood neurodevelopmental disorders and analyzing cell-type specific downstream candidate target genes regulated by Chd8 would be very interesting. Following the reviewer's suggestion, we analyzed gene expression data sets in brain from recent literature identifying genes regulated by Chd8 through transgenic mouse and ChIP-seq analysis. We downloaded Chd8 ChIP-seq data⁵ and selected the genes that have a Chd8 peak

in their promoter (2 kb upstream to 1 kb downstream of the actual TSS). We found that 33 of Chd8 direct targets were indeed differentially expressed in excitatory neurons between P21 and P60 in a cluster-specific manner (**Fig. R5**). We have added this finding as Fig. S5d and provided description in the text (**Page 13**).

Fig. R5. Chd8-bound genes differentially expressed between P21 and P60. Bar plot showing the number of developmentally differential expressed Chd8 target genes in the excitatory neuron cell clusters. Only the clusters with differentially expressed targets are shown.

3. Sex and number of animals used in all studies should be mentioned.

Response: We have described this in detail now as a separate section in the methods (**Page 37**).

4. For P21: semantically: Pg 3: did not, in fact, analyze “through adolescence”, they compared P21 and P60. Especially since P21 is weaning day, these mice have not had “adolescent experience” of new independence as described on pg 10.

Response: We have now replaced the term “through” adolescence, with comparison between “adolescence and adult” to reflect a more accurate description of the experiments and data.

5. The authors should better explain the rationale for elimination of high-mitochondrial-mRNA-load cells.

Response: A high number of mitochondria mRNA is often an indication of dying cells, cell stress or cells with a broken membrane which compromises their viability. From a technical aspect, cells showing a high fraction of mitochondrial gene expression generally lead to an underestimation of the endogenous gene expression resulting to wrong clusters with cells from a mixture of cell types or leading to non-meaningful results when doing differential gene expression analysis. This issue have been investigated in detail in some publications including *Ilicic T et al*⁶ and *Young M.D and Behjati S.*⁷. We have added more explanation in the manuscript (**Page 38**).

6. Pg 6 paragraph 2: there is only one staining experiment shown, but is referred to as an example. If there were other staining experiments utilizing other markers, they should be shown in supplementary figures.

Response: Restricted expression of several predicted markers could be validated from data reported in the Allen Brain Atlas (as shown in Fig. S2 and S4 for excitatory neurons). To provide further proof-of-concept, we co-stained for *Pou3f1*, *Tshz2* and *Foxp2*. While other excitatory markers were not stained, most of them can be visualized in Allen Brain. We have added to the text to clarify this further.

7. *Cartpt* is a diagnostic marker for some interneurons. Yet, only excitatory neurons were analyzed in the cocaine experiments with respect to *Cartpt*; this seems like an interesting piece of information that is missing.

Response: We detected *Cartpt* as a marker of a distinct interneuron subpopulation. This subpopulation co-expressed *Sst*. In fact, the *Sst* neurons could be subdivided into two distinct subtypes based on the expression of *Cartpt* (Fig. 2g,h). However, due to the low abundance of interneurons, few differentially expressed genes could be detected with statistical accuracy, leaving even lower power to make interpretations in downstream analyses (Fig. S8e). Therefore,

this question, although very interesting, has to be reserved in a future studies (potentially by isolation of large numbers of genetically labeled *Cartpt* interneurons).

8. Pg 9, Comparison of VISp and G11.....data is not shown. This should be added to supplementary figures ?

Response: We thank the reviewer for the suggestion. The data is now included as a supplementary table (Supplementary Table 3).

9. Fig. 4f, Please clarify the comparison being made.

Response: This is the comparison of relative expression of epigenetic factors in the different neuronal clusters between P21 and P60. We applied a 1.5 FC cutoff, and indicated downregulated factors in blue and upregulated ones in red. More detailed description is now provided in legend.

10. ASD and ADHD are two very separate clinical diagnoses- They should not be lumped.

Response: We thank the reviewer for the suggestion. We separated ASD and ADHD GWAS candidates and re-did the figures. We updated **Figures 5d, S8c and S8d** accordingly.

11. GO analysis: The authors should clarify methods for GO, add GO number to each graph/function shown. Which GO analysis was run? (Cell component, Biological process, or Molecular function) What is the p-value? Was there any correction of data or p-value? Which statistical parameters were used? Was a software used, or the webpage? When citing pathways that were most significantly changed, what was the p-value?

Response: We thank the reviewer for pointing this out. The functional enrichment analyses were conducted using the commercial Ingenuity Pathway Analysis (IPA) program from Qiagen that maintains an exhaustive and regularly annotated database (<https://www.qiagenbioinformatics.com/products/ingenuity-pathway-analysis/>). Analyses were

run under “Canonical Pathways” and “Disease and Functions” that annotates all the GO terms in IPA. A p-value cutoff of 0.05 was used in reporting pathways and functions that significantly changed. We have now added these information to the “Functional enrichment analysis” section of the Methods.

We would like to thank the reviewers for their constructive comments, which have helped us to improve our manuscript greatly. We hope that they are satisfied with our response.

Sincerely,

Yi Zhang, Ph.D

Professor of Genetics

Harvard Medical School

References

- 1 Mayer, C. *et al.* Developmental diversification of cortical inhibitory interneurons. *Nature* **555**, 457-462, doi:10.1038/nature25999 (2018).
- 2 Francesconi, M. *et al.* Single cell RNA-seq identifies the origins of heterogeneity in efficient cell transdifferentiation and reprogramming. *Elife* **8**, doi:10.7554/eLife.41627 (2019).
- 3 Cameron, C. M., Murugan, M., Choi, J. Y., Engel, E. A. & Witten, I. B. Increased Cocaine Motivation Is Associated with Degraded Spatial and Temporal Representations in IL-NAc Neurons. *Neuron*, doi:10.1016/j.neuron.2019.04.015 (2019).
- 4 Murugan, M. *et al.* Combined Social and Spatial Coding in a Descending Projection from the Prefrontal Cortex. *Cell* **171**, 1663-1677.e1616, doi:10.1016/j.cell.2017.11.002 (2017).
- 5 Katayama, Y. *et al.* CHD8 haploinsufficiency results in autistic-like phenotypes in mice. *Nature* **537**, 675-679, doi:10.1038/nature19357 (2016).
- 6 Ilicic, T. *et al.* Classification of low quality cells from single-cell RNA-seq data. *Genome biology* **17**, 29, doi:10.1186/s13059-016-0888-1 (2016).
- 7 Young, M. D. & Behjati, S. SoupX removes ambient RNA contamination from droplet based single cell RNA sequencing data. *bioRxiv*, 303727, doi:10.1101/303727 (2018).

REVIEWERS' COMMENTS:

Reviewer #1 (Remarks to the Author):

The authors have addressed all of my concerns satisfactorily. I also think they did a great job in addressing the questions from other reviewers. Publication of this manuscript in a timely fashion will be a big help for the neurobiology community.

Reviewer #2 (Remarks to the Author):

The authors have done an excellent job of responding to my previous concerns and I have no additional comments.

Reviewer #3 (Remarks to the Author):

All my previous comments were adequately addressed.

In Supplementary Fig S5, however, only a single cell is shown for each example. For at least one example, could the authors show a lower mag image, depicting 20-50 cells, to portray how uniform is the gene expression change.